# Going Beyond Static: Understanding Shifts with Time-Series Attribution

**Jiashuo Liu**[†,*]**, Nabeel Seedat**[‡]**, Peng Cui**[†] **& Mihaela van der Schaar**[‡]
[†]Tsinghua University, [‡]University of Cambridge
`liujiashuo77@gmail.com, ns741@cam.ac.uk`
`cuip@tsinghua.edu.cn, mv472@cam.ac.uk`

## Abstract

Distribution shifts in time-series data are complex due to temporal dependencies, multivariable interactions, and trend changes. However, robust methods often rely on structural assumptions that lack thorough empirical validation, limiting their practical applicability. In order to support an empirically grounded inductive approach to research, we introduce our **T**ime-**S**eries **S**hift **A**ttribution (TSSA) framework, which analyzes *problem-specific* patterns of distribution shifts. Our framework attributes performance degradation from various types of shifts to each *temporal data property* in a detailed manner, supported by theoretical analysis of unbiasedness and asymptotic properties. Empirical studies in real-world healthcare applications highlight how the TSSA framework enhances the understanding of time-series shifts, facilitating reliable model deployment and driving targeted improvements from both algorithmic and data-centric perspectives.

## 1 Introduction

Machine learning models are increasingly deployed in high-stakes settings such as healthcare with a variety of applications such as early disease screening (Cohn et al., 2003; Soriano et al., 2009) or patient risk assessment (Naghavi et al., 2003; Twetman & Fontana, 2009). While model reliability is vital in such stakes settings, ML model performance often degrades when faced with distribution shifts. This challenge becomes particularly pronounced in time-series data, where unlike static data, the non-stationary properties of time-sereis such as trends, seasonality and inherent temporal dynamics add additional complexity to the nature of these distribution shifts. In particular, we refer to a unique challenge manifested in time series namely "shifts in non-stationary properties", where the patterns themselves evolve over time.

In critical fields such as healthcare, ignoring these temporal distribution shifts can have life-threatening consequences, posing risks to patient safety and care quality. For instance, predictive models trained on patient data prior to a major public health crisis, such as the COVID-19 pandemic, may exhibit significant performance drops when deployed in post-pandemic settings (Roland et al., 2022). For example, during the COVID-19 pandemic, not only did static features like the proportion of high-risk patients change (Ngiam et al., 2023; Singh et al., 2023), but also temporal relationships between vital signs like heart rate and respiratory rate or trends in blood oxygen levels also changed in complex ways over time. Consequently, understanding the changes beyond just static shifts is crucial not just to maintain model performance, but also to ensure patient safety and maintain trust in AI-assisted decision-making systems.

Addressing this problem of understanding distribution shifts in time series remains *underexplored* and particularly challenging. Exisiting approaches (Lu et al., 2023; Liu et al., 2024b) draw on methods similar to those used for general static out-of-distribution (OOD) generalization, such as distributionally robust optimization (DRO) (Sagawa et al.; Duchi & Namkoong, 2021; Liu et al., 2022) and causal invariant learning (Peters et al., 2016; Kuang et al., 2018; Arjovsky et al., 2019). These methods, although theoretically compelling, typically overlook the inherent temporal dynamics of time-series data. Moreover, they frequently rely on structural assumptions about distribution

---

[*]Research conducted while visiting the van der Schaar lab at the University of Cambridge.

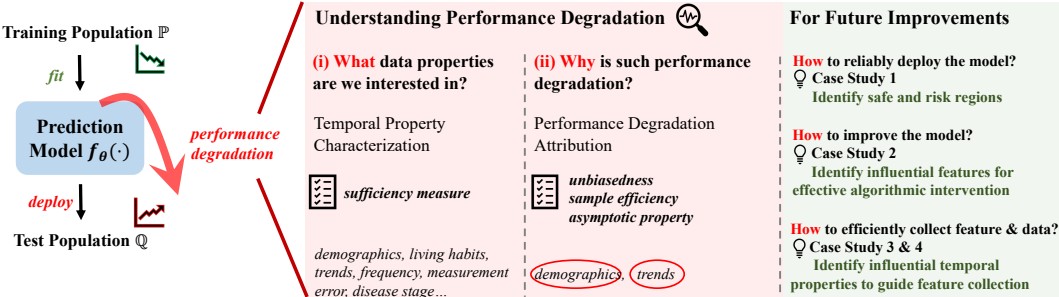

Figure 1: Overview of our TSSA framework.

shifts without rigorous empirical validation, potentially limiting their practical utility (Gulrajani & Lopez-Paz; Yang et al., 2023; Gagnon-Audet et al., 2023; Liu et al., 2023a).

In response to these limitations, we emphasize the importance of adopting an *inductive* approach—one that is grounded in *understanding time-series-specific patterns of distribution shifts*—to effectively address real-world distribution shifts. Rather than arbitrarily applying robust methods or fine-tuning models when performance drops on new test data, we argue for first identifying the specific temporal factors driving the decline. This targeted understanding can guide more effective model adjustments or data collection (Liu et al., 2023a) and enable more judicious model deployment. Recent works (Cai et al., 2023; Feng et al., 2024) propose decomposition approaches to understand and attribute performance drop to different factors. However, they focus on "static" settings and cannot deal with the "shifts in temporal dynamics" problem for time-series data, which necessitates modeling the changes in temporal data properties for time series. Hence, the key question of this work is: How to *attribute* performance degradation between populations to *temporal data properties*?

To tackle this problem, we propose **T**ime-**S**eries **S**hift **A**ttribution (TSSA) framework, which analyzes distribution shifts for time-series data from a data-centric perspective. As shown in Figure 1, our framework involves two parts: **(i)** Define metrics that characterize the behavioral properties of time-series data, along with a *sufficiency measure* that reflects the *optimal predictive power* in estimating the deployed model's prediction error for each sample using these metrics; **(ii)** Attribute performance degradation to each data property in a *detailed* manner. We derive an unbiased doubly robust estimator, incorporating a sample-efficient model architecture for attribution.

> **Contributions:** TSSA is a framework which to the best of our knowledge is the first to address the issue of time-series distribution shift understanding and attribution bringing several contributions: ① **Technically:** we explicitly model and decompose time-series into interpretable properties and link them to shifts using the TSSA framework, providing a doubly robust estimator for a nuanced attribution and understanding of why and how model performance changes across distributions for time-series predictors. ② **Theoretically:** we characterize the unbiasedness and asymptotic properties for the attribution estimator. ③ **Emprically:** we demonstrate through 4 case studies in real-world health-care applications how our TSSA framework enhances the understanding of time-series shifts and informs both reliable model deployment and targeted improvements in ML models as well as data collections.

## 2 PRELIMINARIES

In the context of time-series classification, let $X = [U, V_{1...t}]^T \in \mathcal{X}$ represent the input covariates, where $U \in \mathbb{R}^{d_u}$ denotes static features and $V_{1...t} \in \mathbb{R}^{d_v \times t}$ denotes multi-variate time-series data up to time step $t$. Let $Y \in \mathcal{Y}$ represent the target prediction outcome. Consider a time-series prediction model $f$, which aims to predict the outcome $Y$ based on the covariates $X$. Suppose $f$ is trained on data pairs $(X, Y)$ sampled from a training distribution $\mathbb{P}$. Let $\ell(f(x), y)$ be a loss function quantifying the prediction error, such as cross-entropy loss or 0-1 loss.

In this work, we consider a situation where we observe a performance degradation from $\mathbb{P}$ to a target population $\mathbb{Q}$, i.e. $\mathbb{E}_{\mathbb{Q}}[\ell(f(X), Y)] > \mathbb{E}_{\mathbb{P}}[\ell(f(X), Y)]$. In time-series data, performance degradation can arise from several factors, including changes in the distribution of input variables such as $U$ and $V_{1...t}$, alterations in the temporal properties of the time series $V_{1...t}$, and the presence of missing

variables. Understanding and attributing these shifts is crucial for effectively improving model robustness and ensuring accurate predictions in new settings.

Consider a time-series model designed to predict patient mortality risk based on demographic features and various health metrics monitored in a hospital. The model may have been trained on data collected from the patient population prior to a major public health event but deployed during or after the event, where there could be a significant shift in patient demographics or the emergence of new factors influencing mortality. Additionally, the model may need to be deployed across diverse settings, such as different countries or hospitals. In such cases, we may have sufficient labeled data during the event to evaluate the model, but not enough to train a new one, as investigated in Section 5. Given that collecting new data is extremely costly and could delay patient care, we aim to understand and attribute the performance drop in order to determine in a principled manner the best way to adapt our model or collect the targeted data. Before presenting our methodology, we first discuss the challenges associated with time-series data, and then review the literature on performance drop attribution.

**Challenges in time-series data** Distribution shifts in time-series data are complex, involving both temporal shifts within distributions (e.g., trends, seasonality) and changes between distributions, a challenge we call the "shifts in shifts" problem. Unlike static data, time-series requires addressing these evolving patterns. For instance, pre-pandemic blood oxygen levels in healthy individuals were stable, but during COVID-19, they dropped significantly and continued fluctuating in recovered patients. This highlights the need to consider temporal shift patterns alongside standard shifts.

**Attributing performance drops** Cai et al. (2023) propose decomposing performance drops into two general sources: covariate shifts ($X$) and concept shifts ($Y|X$). Building on this, Feng et al. (2024) employ Shapley values for a more detailed decomposition. However, designed for static data, they cannot be directly applied to time-series data, where, as discussed above, the distribution shifts are much more complicated. Furthermore, since we are only concerned with the contribution of a single data property to the model's performance degradation while all other properties (or features) remain constant, a comprehensive calculation of Shapley values may be unnecessary and less appropriate. Additionally, there are studies that quantify the importance of shifts in individual features within a (partially) known causal or problem structure (Wu et al., 2021; Thams et al., 2022; Zhang et al., 2023). However, these methods rely heavily on expert knowledge, and any misspecification could introduce significant risks in practice.

To the best of our knowledge, our work is the first to analyze detailed shift patterns in time-series data. While extensive research on time-series forecasting addresses temporal shifts *within a single distribution* (Kim et al., 2021; Fan et al., 2023), our work focuses on the more general problem of *shifts in temporal dynamics*. This work primarily aims to understand performance drops in time-series *classification* models. In the following sections, we will introduce metrics to quantify these temporal shifts (Section 3.1) and develop a framework to attribute performance degradation to specific shift patterns (Section 3.2), to address the aforementioned challenges and limitations. More related works on distribution shifts, Shapley Value, and time-series anomaly detection can be found in Appendix A.

## 3 METHOD

In this section, we introduce our **T**ime-**S**eries **S**hift **A**ttribution (TSSA) framework, designed to provide a detailed analysis of model performance degradation in time-series data. The key challenges are two-fold: capturing the temporal properties of time-series data and attributing the overall performance degradation to individual features or properties. Corresponding with these two challenges, as shown in Figure 1, our framework contains two parts: (1) **Temporal Property Characterization**: We define and extract temporal properties, such as global, local and structural properties of the time series, that influence predictive performance. The sufficiency of these metrics is assessed through a novel *Sufficiency Measure*, designed to quantify the extent to which each temporal feature contributes to model performance degradation. (2) **Performance Attribution**: We attribute model performance deterioration to individual temporal properties. This attribution is conducted with a doubly robust estimator, ensuring both *unbiasedness* and *asymptotic consistency* under mild assumptions. We also leverage a shared representation space to efficiently estimate risk models and propensity scores, particularly in scenarios with limited target population data, thereby improving estimation reliability.

## 3.1 DESIDERATA FOR TIME-SERIES METRICS

To systematically capture the complex nature of distribution shifts in time series data, we propose the following desiderata for metrics to attribute performance shifts, which cover various aspects of time series behavior subjected to distribution shift. We categorize these desiderata into four main groups:

**Global Characteristics** Capture the overall, long-term behavior of the time series. These could encompass: (1) Overall Statistics: The overall statistics of a sequence of data, like the average value and standard deviation. (2) Trends: Metrics to capture long-term directional movements in the data. (3) Frequency: Metrics to identify and quantify cyclical patterns or periodic components. (4) Noise level: Metrics to quantify the signal-to-noise ratio. In this work, we introduce **8** metrics to capture the global characteristics, including the *average, max, min, standard deviation values, standardized trend, smoothed trend, maximum frequency*, and *signal-to-noise ratio*.

**Local Dynamics** Assess short-term behaviors and local patterns within the time series. These could encompass: (1) Variability Assessment: Metrics to measure the degree and nature of fluctuations in the time series. (2) Local Non-stationarity: Metrics to detect short-term shifts. (3) Outlier and Anomaly Detection: Metrics to identify outlier or unusual values. In this work, **6** metrics are included to capture the local dynamics, namely *short-term variability, high-frequency energy, normalized Jitter index, relative strength index, KPSS non-stationary test,* and *the number of breakout points*.

**Structural Changes** Identify significant shifts or changes in the underlying data-generating process. These could encompass (1) Metrics to detect abrupt shifts in the statistical properties of the time series. (2) Metrics to capture the shifts in local trends associated with abrupt shifts. We introduce **2** metrics to capture the structural changes, including *the number of change points*, and *the trend variability*.

**Inter-series Relationships** Address the interactions between multiple time series. These could encompass metrics to capture relationships and dependencies between different time series. We introduce the *covariance variability* among multiple time series to capture this.

These desiderata are motivated by the multifaceted nature of distribution shifts in time series data. Global characteristics can reveal shifts in overall patterns, while local dynamics can capture more subtle changes that might be masked in aggregate measures. Structural changes are crucial for identifying significant alterations in the underlying process, and inter-series relationships are essential for understanding shifts in complex, multivariate time series. While not exhaustive, as shown in Table 1, these metrics are drawn from various fields, and represent a comprehensive attempt to instantiate each desideratum, and more metrics can be added in future developments. The detailed definitions can be found in Appendix F.

Table 1: Various metrics to quantify temporal properties from different aspects.

| Temporal Property | Name | Metric | Domain |
|---|---|---|---|
| Global Characteristics | Overall Statistics | Average, Standard Deviation, Max, Min values | Statistics |
| | Standardized Trend | Equation (16) | Statistics |
| | Smoothed Trend | Savitzky-Golay Filter (Savitzky & Golay, 1964) | Analytical Chemistry |
| | Maximum Frequency | Dominant frequency by FFT | Signal Processing |
| | Signal-to-Noise Ratio | Equation (17) | Signal Processing |
| Local Dynamics | Short-Term Variability | Equation (19) | Signal Processing |
| | High-Frequency Energy | Equation (20) | Signal Processing |
| | Normalized Jitter Index | Equation (21) | Signal Processing |
| | Relative Strength Index | Equation (22) | Finance |
| | KPSS Non-Stationary Test | $p$-Value from KPSS Test | Economics |
| | Breakout Points | Equation (18) (Bollinger Bands (Bollinger, 1992)) | Finance |
| Structural Changes | Change Points | PELT (Killick et al., 2012) | Statistics |
| | Trend Variability | Standard deviation of local trends | Statistics |
| Multivariate Interaction | Covariance Variability | Equation (23) | Finance |

With these metrics, we combine the static features $U$ with the metrics of all time-series features $V_{1...t}$, collectively referred to as $\tilde{X}$ in the following sections of this paper.

**Sufficiency Measure** Before moving on to the attribution, one natural question is whether the designed metrics are good enough to capture the temporal properties of time series. Since the ultimate goal is to understand the predictive performance drop, the primary requirement for these metrics is that they should relate to predictive performance. Therefore, we propose a *sufficiency measure* to

evaluate the *optimal predictive power* of the metrics, defined as:

$$\text{Suff.}(\tilde{X}) := \min_{g \in \mathcal{G}} \mathbb{E}\big[Loss(g(\tilde{X}), \ell(f(X), Y))\big], \tag{1}$$

where $Loss(\cdot, \cdot)$ denotes some loss functions (e.g., mean squared error, 0-1 loss) to measure the gap between the *predicted error* and the *real error* of the deployed (and fixed) model $f$, $\mathcal{G}$ can be chosen as any model classes (e.g., neural networks, XGBoost, etc.). This metric measures the optimal power in predicting the deployed model's prediction error for each sample using the data property metrics. The smaller it is, the more predictive the metrics become. Therefore, it can serve as a guideline to measure the quality of the metrics. In Proposition 2, we demonstrate how this sufficiency measure affects the asymptotic properties of our attribution estimation (introduced in Section 3.2). Moreover, in our first case study (Section 5.1), we demonstrate how temporal property metrics can help to effectively identify, in a highly interpretable manner, both the "*safe* region", where the model performs reliably, and the "*risk* region", where it is less reliable. In Appendix D, we discuss more about the utility of our sufficiency measure.

## 3.2   PERFORMANCE DROP ATTRIBUTION

Based on the collection $\tilde{X}$ of static features $U$ and temporal data properties in Table 1, we propose to attribute the performance drop, i.e., $\mathbb{E}_{\mathbb{Q}}[\ell(f(X), Y)] - \mathbb{E}_{\mathbb{P}}[\ell(f(X), Y)]$ to each component in $\tilde{X}$. To assess how a specific data property contributes to the performance drop, inspired by treatment effect estimation, we control for the distribution of all other features except the data property of interest. For instance, if we are interested in understanding the effect of blood pressure on the performance decline, we could first *control* for the distribution of all other features like demographics, blood oxygen level, heart rate, to be the same across both populations. The remaining performance drop, after controlling for other factors, could then be attributed to the effect of blood pressure. Additional demonstrations on the relationship between our TSSA approach and average treatment effect (ATE) estimation can be found at Appendix C.

**Objective of Attribution**   For a specific data property $S$ of interest, let $\tilde{X}_{\backslash\{S\}}$ denote all other properties/features in $\tilde{X}$, abbreviated as $\tilde{X}_{-S}$ in the remainder of this work. To "control" for the effects of $\tilde{X}_{-S}$, we define the conditional risks under distributions $\mathbb{P}$ and $\mathbb{Q}$ as follows:

$$R_{\mathbb{P}}(\tilde{X}_{-S}) := \mathbb{E}_{\mathbb{P}}[\ell(f(X), Y) \mid \tilde{X}_{-S}], \quad R_{\mathbb{Q}}(\tilde{X}_{-S}) := \mathbb{E}_{\mathbb{Q}}[\ell(f(X), Y) \mid \tilde{X}_{-S}], \tag{2}$$

where $\ell(f(X), Y)$ denotes the deployed model's *original* prediction error on the sample $(X, Y)$. Note that when measuring prediction error $\ell(f(X), Y)$, we use the *original samples* rather than the extracted temporal properties. Building on this, we define the attribution score of feature $S$ as:

$$\text{Attr.}(S) := \mathbb{E}[R_{\mathbb{Q}}(\tilde{X}_{-S}) - R_{\mathbb{P}}(\tilde{X}_{-S})], \tag{3}$$

which quantifies the performance gap between $\mathbb{Q}$ and $\mathbb{P}$, while holding the marginal distribution of all features except the feature of interest, $\tilde{X}_{-S}$, constant. From a distribution shift perspective, $\text{Attr.}(S)$ quantifies the performance drop introduced by $(Y, S)|\tilde{X}_{\backslash\{S\}}$-shifts between $\mathbb{P}$ and $\mathbb{Q}$.

Furthermore, in cases where we do not isolate a specific feature (i.e. feature of interest is set to the empty set), i.e., $\text{Attr.}(\emptyset)$, the attribution score captures the "systematic" (and unavoidable) difference between $\mathbb{P}$ and $\mathbb{Q}$, potentially caused by missing information. For example, in predicting patient risk, individuals from two populations may have similar health indices yet experience vastly different outcomes due to missing information, such as differing living habits across regions or the absence of key health metrics in the records. Note that this term reduces to the concept of "$Y|X$-shift" introduced by Cai et al. (2023). To effectively estimate Equation (3), we propose a doubly robust estimator for our objective Equation (3) as it is resilient to misspecification, ensuring reliable attribution even when some assumptions might be violated.

**Doubly Robust Estimation for Attribution**   Consider the original data $(U^i, V^i_{1...t}, Y^i)^{n_P}_{i=1} \sim \mathbb{P}$ and $(U^j, V^j_{1...t}, Y^j)^{n_Q}_{j=1} \sim \mathbb{Q}$, we first calculate the temporal properties in Table 1 for $V_{1...t}$, and convert the data into $(\tilde{X}^i, Y^i)^{n_P}_{i=1} \sim \mathbb{P}$ and $(\tilde{X}^j, Y^j)^{n_Q}_{j=1} \sim \mathbb{Q}$. To estimate the attribution, i.e. Equation (3), we first learn two predictors $\hat{\mu}_{\mathbb{P}}(\cdot)$, $\hat{\mu}_{\mathbb{Q}}(\cdot)$ to approximate the conditional risk function $R_{\mathbb{P}}(\cdot)$, $R_{\mathbb{Q}}(\cdot)$ in Equation (2) respectively from the observation data. Then we fit a domain classifier $\hat{\pi}(\cdot)$ as (propensity score estimator):

$$\hat{\pi}(x_{-S}) \approx \Big\{ \pi(x_{-S}) := \Pr\big(x_{-S} \text{ from } \mathbb{Q} \big| \tilde{X}_{-S} = x_{-S}\big) \Big\}. \tag{4}$$

We use non-parametric models for $R_\mathbb{P}(\cdot), R_\mathbb{Q}(\cdot), \pi(\cdot)$, ensuring flexibility in learning the relationships between variables without relying on strong parametric assumptions. Throughout the theoretical analysis in this paper, we consider generic nonparametric regression estimators, and all strategies could be used directly with different ML models, e.g. tree-ensembles. Then we formulate the Augmented IPW (AIPW) (Robins et al., 1994) estimator as:

$$\widehat{\text{Attr.}}(S) = \frac{1}{n_P + n_Q} \bigg( \sum_{i=1}^{n_P+n_Q} \left( \hat{\mu}_\mathbb{Q}(\tilde{X}^i_{-S}) - \hat{\mu}_\mathbb{P}(\tilde{X}^i_{-S}) \right) +$$
$$\sum_{j=1}^{n_Q} \frac{R_\mathbb{Q}(\tilde{X}^j_{-S}) - \hat{\mu}_\mathbb{Q}(\tilde{X}^j_{-S})}{\pi(\tilde{X}^j_{-S})} - \sum_{i=1}^{n_P} \frac{R_\mathbb{P}(\tilde{X}^i_{-S}) - \hat{\mu}_\mathbb{P}(\tilde{X}^i_{-S})}{1 - \pi(\tilde{X}^i_{-S})} \bigg), \tag{5}$$

where $R_\mathbb{Q}(\tilde{X}^j_{-S})$ denotes the ground-truth value on sample $\tilde{X}^j_{-S}$ that can be calculated from our observation data (the same for $R_\mathbb{P}(\tilde{X}^i_{-S})$). We also demonstrate the compatibility of our attribution with Shapley Value in Appendix B.

This aforementioned formulation ensures double robustness, meaning that our attribution remains consistent and unbiased as long as either the conditional risk predictors or the propensity score estimator is correctly specified. With the propensity score estimator $\hat{\pi}(\cdot)$ in use, we rely on the overlap assumption between $\mathbb{P}$ and $\mathbb{Q}$, a common requirement in causal inference (Wager, 2020). There are several approaches to address potential non-overlap in practice (Cai et al., 2023). In this work, we adopt a simpler strategy to mitigate the effects of non-overlap by excluding samples with propensity scores that are extremely close to 0 or 1. Furthermore, in Section 4, we prove the *unconfoundedness*, *unbiasedness*, and the *asymptotic properties* of our estimator in Equation (5).

**Sample-Efficient Estimation of $\hat{\mu}_\mathbb{P}(\cdot), \hat{\mu}_\mathbb{Q}(\cdot),$ and $\hat{\pi}(\cdot)$ with Neural Networks**   In non-parametric estimation, we generally have sufficient samples for $\hat{\mu}_\mathbb{P}(\cdot)$ (from the training population $\mathbb{P}$). However, in real-world applications, data from the target population Q is often scarce, making it challenging to estimate $\hat{\mu}_\mathbb{Q}(\cdot)$ and $\hat{\pi}(\cdot)$ accurately. To mitigate this, given the functions share the same input $\tilde{X}_{-S}$, we propose to learn a shared representation space, thereby sharing information between the two populations, and improving the sample efficiency of our estimates. Motivated by Shi et al. (2019), we adopt the model architecture as shown in Figure 2. Note that more advanced architectures (Curth & Van der Schaar, 2021) can be adopted here, and this is not the focus of this work. Through this model, we share information between samples from $\mathbb{P}$ and $\mathbb{Q}$ to learn the representation space, which helps for the estimation of $\hat{\mu}_\mathbb{Q}(\cdot)$ and $\hat{\pi}(\cdot)$. Specifically, the loss function is:

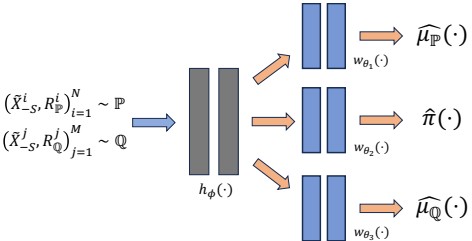

Figure 2: Sample-efficient estimation with neural networks. $R^i_\mathbb{P}$ denotes the conditional risk associated with $i$-th sample drawn from $\mathbb{P}$ (similar for $R^j_\mathbb{Q}$). $h_\phi(\cdot)$ represents the shared representation learner, and $w_{\theta_1}, w_{\theta_2}, w_{\theta_3}$ denote three separate predictors.

$$\min_{\phi, \theta_1, \theta_2, \theta_3} \underbrace{\sum_{i=1}^{n_P} (w_{\theta_1}(h_\phi(\tilde{X}^i_{-S})) - Y^i)^2}_{\text{for } \hat{\mu}_\mathbb{P}(\cdot)} + \underbrace{\sum_{j=1}^{n_Q} (w_{\theta_3}(h_\phi(\tilde{X}^j_{-S})) - Y^j)^2}_{\text{for } \hat{\mu}_\mathbb{Q}(\cdot)} + \underbrace{\sum_{i=1}^{n_P+n_Q} \ell_{\text{CE}}(w_{\theta_2}(h_\phi(\tilde{X}^i_{-S})), Y^i)}_{\text{for } \hat{\pi}(\cdot)},$$

where $\ell_{\text{CE}}(\cdot, \cdot)$ denotes the cross-entropy loss.

## 4 THEORETICAL ANALYSIS

In this section, we characterize the unbiasedness and asymptotic properties for our estimation.

**Proposition 1** (Unconfoundedness & Unbiasedness). *Denote $T \in \{0, 1\}$ as an indicator variable:*

$$T = 0, \quad \text{if } \tilde{X}_{-S} \text{ is from } \mathbb{P}; \quad T = 1, \quad \text{if } \tilde{X}_{-S} \text{ is from } \mathbb{Q}, \tag{6}$$

*which can be likened to a treatment variable. For any attribute $S \in \tilde{X}$, we have:*

$$\left\{ R_\mathbb{P}(\tilde{X}_{-S}), R_\mathbb{Q}(\tilde{X}_{-S}) \right\} \perp\!\!\!\perp T \mid \tilde{X}_{-S}. \tag{7}$$

*Based on this, assume that for all $\tilde{X}_{-S}$, the overlap assumption holds, i.e., $0 < \pi(\tilde{X}_{-S}) < 1$, then the estimator in Equation (3) is consistent if either the $\hat{\mu}_{\mathbb{P}}(\cdot), \hat{\mu}_{\mathbb{Q}}(\cdot)$ are consistent or $\hat{\pi}(\cdot)$ is consistent.*

Proof can be found in Appendix H. In addition to the unconfoundedness and unbiasedness, based on Wager (2020), we *quantify* the consistency between our estimator and the *oracle* one as follows.

**Proposition 2** (Consistency). *Consider the* oracle *estimator that uses the true* $\mu_{\mathbb{P}}(\cdot) = R_{\mathbb{P}}(\cdot)$, $\mu_{\mathbb{Q}}(\cdot) = R_{\mathbb{Q}}(\cdot)$, *and* $\pi(\cdot)$ *rather than the estimates thereof:*

$$\widehat{Attr.}^{\star}(S) = \frac{1}{n_P + n_Q} \sum_{i=1}^{n_P + n_Q} \left( R_{\mathbb{Q}}(\tilde{X}^i_{-S}) - R_{\mathbb{P}}(\tilde{X}^i_{-S}) \right), \tag{8}$$

*Assume that all samples are independent, and for all* $\tilde{X}_{-S}$, *the overlap assumption holds, then:*

$$|\widehat{Attr.}(S) - \widehat{Attr.}^{\star}(S)| = \mathcal{O}_P\left( \underbrace{\max_{\tau \in \{\mathbb{P}, \mathbb{Q}\}} \mathbb{E}\left[ (\hat{\mu}_\tau(\tilde{X}_{-S}) - R_\tau(\tilde{X}_{-S}))^2 \right]^{\frac{1}{2}}}_{\text{sufficiency measure in Equation (1)}} \mathbb{E}\left[ (\hat{\pi}(\tilde{X}_{-S}) - \pi(\tilde{X}_{-S}))^2 \right]^{\frac{1}{2}} \right).$$

*Proof can be found in Appendix H.*

**Remark 1.** *Proposition 2 quantifies how closely our estimator approximates the oracle. We can make the following observations: (i) The first term,* $\max_{\tau \in \{\mathbb{P}, \mathbb{Q}\}} \mathbb{E}\left[ (\hat{\mu}_\tau(\tilde{X}_{-S}) - R_\tau(\tilde{X}_{-S}))^2 \right]$, *characterizes the prediction error of* $\hat{\mu}_{\mathbb{P}}(\cdot)$ *and* $\hat{\mu}_{\mathbb{Q}}(\cdot)$*. This term directly corresponds to the* sufficiency measure *in Equation (1), which represents the objective of our designed temporal data properties, thereby linking the two stages of our framework. (ii) For any* $o_P(n^{-1/4})$*-consistent machine learning methods to estimate* $\hat{\mu}_{\mathbb{P}}, \hat{\mu}_{\mathbb{Q}}, \hat{\pi}$, *i.e.*

$$\max_{\tau \in \{\mathbb{P}, \mathbb{Q}\}} \mathbb{E}\left[ (\hat{\mu}_\tau(\tilde{X}_{-S}) - R_\tau(\tilde{X}_{-S}))^2 \right]^{\frac{1}{2}} \mathbb{E}\left[ (\hat{\pi}(\tilde{X}_{-S}) - \pi(\tilde{X}_{-S}))^2 \right]^{\frac{1}{2}} \ll \frac{1}{\sqrt{n}}, \quad \text{then we have} \tag{9}$$

$$\sqrt{n}\left( \widehat{Attr.}(S) - \widehat{Attr.}^{\star}(S) \right) \to_p 0. \tag{10}$$

## 5 EXPERIMENTS

We design 4 case studies to comprehensively demonstrate the effectiveness and usage of our TSSA framework, including different real-world shift patterns and prediction tasks. Through our experiments, we use the Medical Information Mart for Intensive Care (MIMIC) (Johnson et al., 2016) dataset. It is representative of *complex real-world* medical time series and contains 23,100 patients from which 9 static demographic features (such as age, gender, admission type etc) and 53 time-series health indexes (such as blood pressure, Braden mobility, temperature etc) have been measured.

### 5.1 CASE STUDY 1: TEMPORAL PROPERTIES GUIDING RELIABLE MODEL DEPLOYMENT

In the first case study, we demonstrate the importance of our time-series data properties (Appendix F) through the interpretable guidance for model safe deployment. Consider a typical intensive-care scenario where the classifier, $f_\theta(\cdot)$, is trained on historical data but is deployed across different patients and at various stages of their care. Since different patients are likely to exhibit varying feature patterns, it is challenging for a single model to perform consistently well across all incoming patients. Therefore, in high-stakes scenarios like this, it is crucial to *identify in advance* the types of data where the model performs reliably and clinicians can trust its predictions—referred to as the *safe region*. Similarly, it is important to recognize the data patterns where the model performs poorly and should not be relied upon, referred to as the *risk region*.

**Experiment Setup** The task is to predict patient mortality based on 24-hour recordings. We follow the standard design outlined by Jarrett et al., randomly splitting the patients in the MIMIC-III dataset into a training set (18,490 patients, $\mathbb{P}$) and a test set (4,610 patients, $\mathbb{Q}$), ensuring no patient overlap between the two sets. For the validation set, we use the same patients as in the training set but select different time segments for their time-series features, denoted as $\mathbb{P}_{\text{val}}$. We train a Transformer model, $f_\theta(\cdot)$, on the training set, perform region analysis based on the validation data, and use the test set to verify the effectiveness of the identified regions.

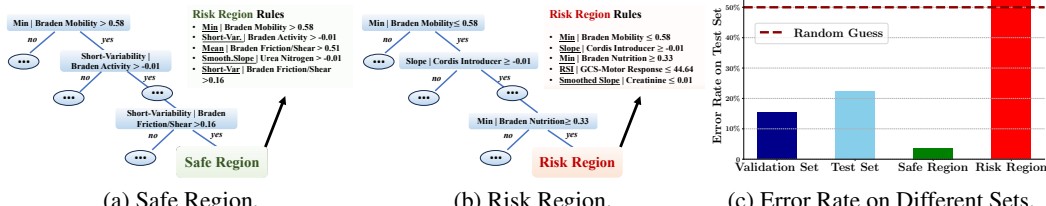

(a) Safe Region.  (b) Risk Region.  (c) Error Rate on Different Sets.

Figure 3: Region Analysis. (a)-(b): Visualizations of the safe and risk regions, defined by interpretable decision rules based on the extracted temporal properties. (c) The error rates on the validation set, the entire test set, the safe region, and the risk region. Error rates in the risk region exceed overall test error, while error in the safe region is much lower than overall test error, thereby guiding safe model deployment by identifying and avoiding usage for patients in the risk region. Note that the regions are learned on validation set $\mathbb{P}_{val}$ with no access of the test set in advance.

**Methodology in Region Analysis** Given the high-stakes nature of the task, these regions must be highly interpretable, rather than relying on opaque, non-interpretable parametric models. Inspired by Lim et al. (2021); Liu et al. (2023a), we fit a decision tree model, $h(\tilde{X})$ on the validation set ($\mathbb{P}_{val}$), to predict the prediction error $\ell(f_\theta(X), Y)$ of the trained model $f_\theta(\cdot)$, using the extracted data properties and static demographic features $\tilde{X}$ (Appendix F). The decision tree partitions the data into distinct regions, each defined by interpretable decision rules and corresponding to a unique leaf node. By analyzing the samples within each leaf, we can identify regions with the highest and lowest risk, as the label assigned by the decision tree represents the original prediction error.

**Analysis** In Figure 3a and Figure 3b, we visualize the safe and risk regions identified based on temporal properties, defined by interpretable and easy-to-understand decision rules. To demonstrate how this guides safe model deployment, we calculate the error rates for the entire test set, the safe region, and the risk region separately. From Figure 3c, while the overall error rate on the test set is relatively reasonable (22.4%), the error rate in the risk region is much higher (52.0%), nearing the level of random guessing. In contrast, the error rate in the safe region is significantly lower (3.7%). Our region analysis offers not only interpretability by design, but also a tool for reliable model deployment. Clinicians can confidently apply the model on patients in the safe region while avoiding use for patients in the risk region, thus ensuring reliable model deployment, which is important from the perspective of trustworthy ML in high stakes settings like healthcare. Further results in Appendix G.3 reinforce the superiority of our temporal-property-based region analysis.

**Take-away 1:** Temporal properties enable accurate and interpretable identification of safe and risk regions, ensuring the reliable and smart deployment of ML models in critical care.

## 5.2 CASE STUDY 2: AGE SHIFTS IN MORTALITY RISK PREDICTION

In real-world healthcare settings, shifts in population distribution are commonly observed. For instance, significant variations in patient age distributions often arise across different hospitals and regions. To rigorously assess the effectiveness of our attribution method, we design scenarios reflecting these variations in patient age on MIMIC-III, and verify that our framework attributes the observed performance degradation to the appropriate features or properties.

**Experiment Setup: Shifts in Patient Age** The task is to predict patient mortality based on 24-hour recordings. We consider a data collection process that oversamples patients under the age of 65, where the average age of the patients in training is 57, while in the test set, the average age is 77. The training set contains 11,476 patients (training distribution $\mathbb{P}$), and the test set contains 6,408 new (but older) patients (target distribution $\mathbb{Q}$). For the validation data, similar with Case Study 1 (Section 5.1), we choose the same patients as in the training set but select different time segments for their time-series features. We train a Transformer model $f_\theta(\cdot)$ on $\mathbb{P}$ and evaluate it on $\mathbb{Q}$, where we observe a performance drop of 13.9pp on accuracy (from 87.7% to 73.8%), and 8.8pp on Macro-F1 score (from 72.5% to 63.7%). In the subsequent analysis, we keep the model $f_\theta(\cdot)$ fixed during evaluation and apply our TSSA framework to attribute its performance drop to various features. More details can be found in Appendix G.6.

**Analysis** We begin by presenting the feature attribution results in Figure 4a, which show the average attribution scores along with standard deviation errors from 10 random runs. The findings reveal that the top features identified by our framework are predominantly demographic variables (yellow bars), with the "Age" feature correctly attributed as the most influential to the performance

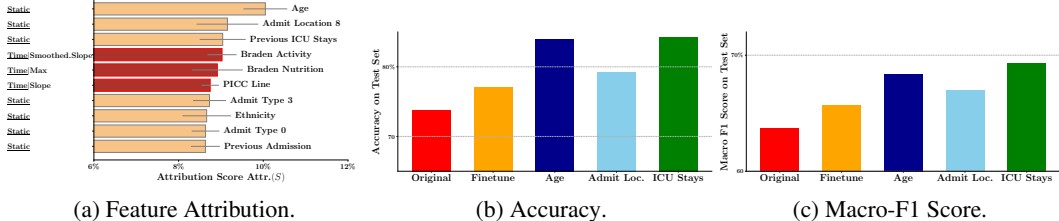

| (a) Feature Attribution. | (b) Accuracy. | (c) Macro-F1 Score. |

Figure 4: Results for Case Study 2: (a) We visualize the top 10 most important features, which are mainly demographic features (yellow bars), with age being the most prominent. This corresponds with our "prior" knowledge and verify that our framework attributes to the appropriate features. (b)-(c) We do some balancing the top three features respectively during the training phase and plot the accuracy and Macro-F1 score for each model. The results show a significant improvement in performance over original model and simple finetuning when we intervene on these top features.

drop. Other key features, such as Admission Location (e.g., emergency room, referral, transfer from other hospitals) and Previous ICU Stay Duration, have a relatively strong correlation with the Age feature. This aligns well with our problem setting, where we perform oversampling based on the "Age" feature. Furthermore, can we use the attribution to guide actions to remedy model performance. For each of the top 3 identified features (i.e., Age, Admit Location, and Previous ICU Stays), we apply *simple* balancing by reweighting the data based solely on the inverse density ratio of that specific feature to achieve a uniform distribution. As shown in Figure 4b and Figure 4c, we observe *significant* improvements in both test accuracy and Macro-F1 score. This demonstrates that the top features identified by our framework directly influence the performance drop observed between the training and test patients. Additionally, this highlights how our attribution results can inspire further algorithmic interventions. In this case study, this inductive approach shows that even simple algorithmic modifications can lead to significant improvements.

**Take-away 2:** Our attribution framework accurately attributes the performance drop to relevant demographic features and inspires straightforward yet effective algorithmic interventions, such as data balancing (as done in our intervention), and targeted data augmentation.

### 5.3 CASE STUDY 3: PREEMPTIVE DIAGNOSIS UNDER TEMPORAL SHIFTS

Beyond shifts in patient age, temporal variations in patients' conditions represent another prevalent type of shift in healthcare data. In preemptive diagnostic scenarios, our target is for the model to detect mortality risks at the earliest possible stage (Filippin et al., 2015). This proactive approach not only enhances patient outcomes by facilitating timely interventions but also underscores the importance of adapting models to account for the dynamic nature of patient health over time. In this case study, we explore how our attribution framework can offer valuable insights for this.

**Experiment Setup: Preemptive Diagnosis under Temporal Shifts** To evaluate reliability in the early detection of mortality risk, we examine the temporal shifts between the training and test patients. Specifically, for the training set $\mathbb{P}$, we utilize the last 24-hour time segments for all time-series features, while for the test set $\mathbb{Q}$, we select the first 24-hour time segments for all features. This setup allows us to assess whether the model can effectively withstand these temporal shifts and accurately identify patients at high risk of mortality in the early stage. We train a Transformer model $f_\theta(\cdot)$ on $\mathbb{P}$, which comprises 12,574 patients, and validate it on an additional 5,547 patients. To control for other shifts, we use the same set of patients for both the validation and test sets $\mathbb{Q}$; the only difference lies in the time segments used: the last 24 hours for validation and the first 24 hours for testing. In this case, we observe a performance drop of 13.8pp in accuracy (from 90.0% to 76.2%) and 24.8pp in Macro-F1 score (from 64.1% to 39.3%), highlighting the necessity to investigate the reasons behind this significant decline. Then for the subsequent analysis, we keep the model $f_\theta(\cdot)$ fixed during evaluation and apply our TSSA framework to attribute its performance drop to various features.

**Analysis** In Figure 5a, we present the average attribution scores along with standard deviation errors from 10 random runs. In contrast to Figure 4a, which displays attribution based on age shifts, the prominent features in this analysis are primarily temporal properties, including the smoothed slope, high-frequency energy, and break points. For instance, for the top feature——the smoothed slope of Braden Activity——we randomly selected ten patients and visualized the last 24-hour (blue) and the first 24-hour (red) time series for Braden Activity in Figure 5b. Here, we observe that the trends, as indicated by the smoothed slope metric (top feature), and break points (sixth feature) of this time series differ significantly. As demonstrate by Valiani et al. (2017), mobility status during

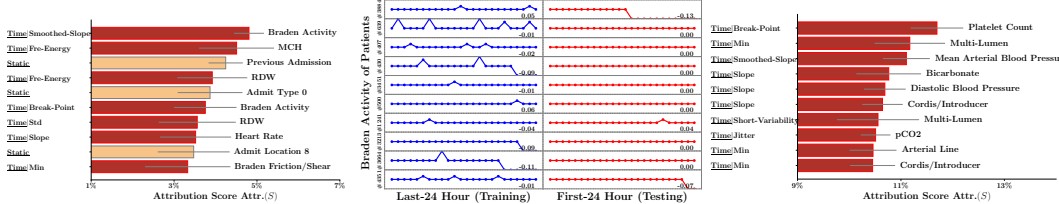

(a) Attribution (Case Study 3).  (b) Braden Activity Curve.  (c) Attribution (Case Study 4).

Figure 5: Results of Case Study 3&4: (a) We visualize the top 10 most important features for Case Study 3. We can see that the performance drop is primarily driven by extracted temporal properties. (b) We randomly select 10 patients to compare the curves of the top feature (Braden Activity). A significant difference in both the slope and breakpoints is observed. (c) We visualize the top 10 most important features for Case Study 4, where they are all temporal properties.

hospitalization provides substantial prognostic value in hospitalized older adults. The Braden Activity score could be an efficient and valuable source of information about mobility status for targeting post-hospital care of older adults. Our attribution results provide valuable insights for clinicians. First, it is essential to monitor changes in the Braden Activity feature closely. Second, as illustrated on the right side of Figure 5b, many of these changes may go unrecorded or unmeasured during the early stages of patient care. To enable timely detection, clinicians should prioritize the assessment of this feature from the outset. Overall, these findings underscore the potential of our results to enhance clinical practices and improve feature collection.

**Take-away 3:** Our attribution framework provides valuable insights to guide ML engineers and clinicians, like implementing timely and effective feature monitoring and collection.

### 5.4    Case Study 4: Ventilator Prediction

We focus on predicting the use of mechanical ventilation in intensive care——a procedure that is both invasive and uncomfortable, requiring the induction of an artificial coma and carrying a significant risk of mortality. Accurate predictions are crucial, as errors can lead to serious consequences.

**Experiment Setup**  The task is to predict whether a patient requires mechanical ventilation. We consider a realistic scenario in which a model trained on historical data (12,574 patients, first 24-hour time series, denoted as $\mathbb{P}$) must be deployed for new patients and future time segments (an additional 5,547 patients, *second* 24-hour time series, denoted as $\mathbb{Q}$). This scenario accounts for shifts in both demographics (across different patients) and temporal factors (across various time periods, such as the first day versus the second day). We train a Transformer model, and observe a performance drop of 12.4pp in accuracy (from 93.7% to 81.3%) and 10.1pp in Macro-F1 score (from 62.4% to 52.3%), highlighting the necessity to investigate the reasons behind this significant decline.

**Analysis**  We present the feature attribution results in Figure 5c. Notably, all top 10 features are temporal properties, indicating that the expected demographic shifts among patients are *unexpectedly minor*. To further explore this phenomenon, we exclude temporal shifts and build a new test set denoted as $\mathbb{Q}'$ where the time-series features are also derived from the first 24 hours (the same as $\mathbb{P}$). After evaluation, the drop from $\mathbb{P}$ to $\mathbb{Q}'$ is only 0.6pp in accuracy and 0.3pp in Macro-F1 score. Thus, this drop can be primarily attributed to demographic shifts among different patients. This further supports our attribution results that indicate minor shifts among patients in this scenario. Furthermore, our identified top features are closely related to the ventilator prediction. For example, Platelet Count (Top-1) is recognized as a significant prognostic marker in intensive care (Mackay et al., 2010; Ilban & Simsek, 2023). Therefore, our attribution results offer valuable insights for ensuring reliable deployment and aiding clinicians in making more informed decisions.

**Take-away 4:** Our attribution framework effectively identifies the primary sources of distributional shifts, providing actionable insights for guiding subsequent algorithmic interventions.

## 6    Conclusion

This paper presented the Time-Series Shift Attribution (TSSA) framework, which effectively attributes performance degradation due to distribution shifts in time-series data, with a focus on healthcare applications. Our empirical and theoretical results demonstrate its potential to enhance model reliability and inspire both algorithmic and data-centric interventions in the future.

## ACKNOWLEDGEMENTS

The authors are grateful to the anonymous ICLR reviewers for their useful comments and feedback. Peng Cui is supported by NSFC (No. 62425206, 62141607) and Beijing Municipal Science and Technology Project (No. Z241100004224009). NS is funded by the Cystic Fibrosis Trust.

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

## A    RELATED WORK

Besides the works introduced in Section 2, we will briefly review other methods related to our topic in this section.

**Distribution Shifts**    To tackle the challenge of distribution shifts (Liu et al., 2021b), several methods have been proposed, with distributionally robust optimization (DRO) (Sagawa et al.; Duchi & Namkoong, 2021; Duchi et al., 2023; Liu et al., 2023a) and invariant learning (Arjovsky et al., 2019; Koyama & Yamaguchi, 2020; Liu et al., 2021a; 2024a) being among the most prominent approaches. DRO techniques involve perturbing the data distribution within a predefined uncertainty set and optimizing for the worst-case prediction risk. However, these uncertainty sets are often selected based on theoretical considerations rather than empirical evidence, which can lead to overly pessimistic outcomes in practice (Liu et al., 2023a). Invariant learning methods, on the other hand, aim to discover representations that maintain a consistent relationship with the outcome variable across different domains. Nevertheless, these methods depend heavily on the availability of high-quality environments (Liu et al., 2021a), and the assumption of invariance may not hold in real-world settings. For instance, unobserved confounders or missing variables—common in practical scenarios—can prevent the existence of a robust invariant representation. Our work addresses the distribution shift problem through an inductive approach, first aiming to understand the shift patterns. This understanding can then inform targeted interventions from both algorithmic and data-centric perspectives.

**Time-Series Anomaly Detection**    In time-series anomaly detection, several works have addressed concept drifts over time, developing both supervised and unsupervised methods (Le & Papotti, 2020; Braei & Wagner, 2020; Liu et al., 2023b). Our approach differs from these methods in several key ways: (i) our goal is to understand why a model's performance declines in a prediction task by attributing this drop to specific properties of interest, whereas anomaly detection methods focus primarily on reliably detecting breakpoints; (ii) the shifts considered in time-series anomaly detection are typically associated with breakpoints, whereas our work covers a broader range of temporal properties (as shown in Table 1). Additionally, our attribution framework can be applied to diagnose performance drops in anomaly detection methods as well.

Furthermore, in the following Appendix B, we demonstrate in detail the related works with Shapley Value (Lundberg & Lee, 2017), as well as the compatibility of our approach with Shapley Value.

## B    SHAPLEY VALUE

One typical attribution method is the SHapley Additive exPlanations (SHAP) (Lundberg & Lee, 2017), which uses cooperative game theory to compute explanations of model predictions. However, our work differs from previous ones in the following aspects:

- **Different goals**: Previous works focus on attributing *model predictions* to input features, whereas our work aims to understand the *performance degradation* of the model when transferring from one distribution to another.

- **Different philosophies**: Shapley values compute the *average* contribution of a feature across all possible subsets of features. In contrast, our approach focuses on the impact of a single data property on the model's performance degradation, while keeping *all other properties (or features) constant*. As a result, a full calculation of Shapley values may be unnecessary and less suitable for our objectives.

- **Different settings**: Previous works typically address static settings, where models can be refit with any subset of features. Our work, however, deals with time-series data, where each time series involves multiple temporal properties, making it infeasible to "remove" one property and refit the model to estimate SHAP values.

Furthermore, our proposed attribution method is compatible with SHAP values, as our attribution score can be integrated into SHAP as the "effect" function. In the following, we would like to first demonstrate why we currently do not use Shapley Value, and then illustrate how our approach can be integrated with Shapley Value.

**Why we do not use Shapley Value**   One typical attribution method is the SHapley Additive exPlanations (SHAP) (Lundberg & Lee, 2017), which uses cooperative game theory to compute explanations of model predictions. However, our work differs from previous ones in the following aspects:

- **Different goals**: Previous works focus on attributing *model predictions* to input features, whereas our work aims to understand the *performance degradation* of the model when transferring from one distribution to another.

- **Different philosophies**: Shapley values compute the *average* contribution of a feature across all possible subsets of features. In contrast, our approach focuses on the impact of a single data property on the model's performance degradation, while keeping *all other properties (or features) constant*. As a result, a full calculation of Shapley values may be unnecessary and less suitable for our objectives.

- **Different settings**: Previous works typically address static settings, where models can be refit with any subset of features. Our work, however, deals with time-series data, where each time series involves multiple temporal properties, making it infeasible to "remove" one property and refit the model to estimate SHAP values.

**Compatibility with Shapley Value**   Furthermore, our proposed attribution method is compatible with SHAP values, as our attribution score can be integrated into SHAP as the "effect" function.

First, recall the definition of Shapley Value: for a player $i$, the Shapley value $\phi_i(v)$ is calculated as:

$$\phi_i(v) = \sum_{\mathcal{S} \subseteq \mathcal{N} \setminus \{i\}} \frac{|\mathcal{S}|!(|\mathcal{N}| - |\mathcal{S}| - 1)!}{|\mathcal{N}|!} \left(v(\mathcal{S} \cup \{i\}) - v(\mathcal{S})\right) \tag{11}$$

Where $\mathcal{N}$ is the set of all players, $\mathcal{S}$ is a subset of players that does not include $i$, $|S|$ and $|\mathcal{N}|$ are the sizes of sets $\mathcal{S}$ and $\mathcal{N}$, respectively, $v(\mathcal{S})$ is the value function for coalition $\mathcal{S}$, representing the payoff that the players in $\mathcal{S}$ can generate. $v(\mathcal{S} \cup \{i\}) - v(\mathcal{S})$ is the marginal contribution of player $i$ when joining coalition $\mathcal{S}$.

Second, we demonstrate how our TSSA approach can be integrated with Shapley Value. Denote the attribution results of feature $S$ given by our original approach as $\widehat{\text{Attr.}}(S)$. Then we can define the Shapley Value as:

$$\widehat{\text{SHAP.Attr.}}(S) = \sum_{\mathcal{V} \subseteq X_{-S}} \frac{|\mathcal{V}|!(d - |\mathcal{V}| - 1)!}{d!} (\widehat{\text{Attr.}}(\mathcal{V} \cup S) - \widehat{\text{Attr.}}(\mathcal{V})), \tag{12}$$

where $d$ denotes the number of extracted features, $X_{-S}$ denotes the set of all features except $S$. Therefore, our TSSA approach is compatible with Shapley Value, which we leave as a promising way of future extension of this work.

## C   DISCUSSION ON THE RELATIONSHIP WITH AVERAGE TREATMENT EFFECT

**Relationship with Average Treatment Effect (ATE)**   First, we would like to clarify that our TSSA is an attribution approach, which is not designed to estimate ATE or solve causal problems. And we only "interpret" our objective function as a special kind of ATE. That is, denote $T \in \{0, 1\}$ as an indicator variable:

$$T = 0, \quad \text{if } \tilde{X}_{-S} \text{ is from } \mathbb{P}; \quad T = 1, \quad \text{if } \tilde{X}_{-S} \text{ is from } \mathbb{Q}, \tag{13}$$

which can be likened to a *treatment variable*. And our attribution objective can then be rewritten as:

$$\text{Attr.}(S) = \mathbb{E}[R(\tilde{X}_{-S}, T = 1) - R(\tilde{X}_{-S}, T = 0)], \qquad (14)$$

where $R(\cdot, T = 1)$ denotes $R_{\mathbb{Q}}(\cdot)$, and $R(\cdot, T = 0)$ denotes $R_{\mathbb{P}}(\cdot)$ that can be viewed as two *outcome function* (like in causal literature). Therefore, when studying the effect of one feature to the performance drop, our attribution approach controls all the other attributes (to be identical between group $T = 1$ and group $T = 0$), and then calculate its effect. Note that we are *not* studying original ATE estimation problems.

**Theoretical Analysis Beyond ATE**    Unlike in causal inference literature, where verifying the unconfoundedness assumption can be challenging, our problem formulation enables us to *prove* the unconfoundedness directly in Proposition 1, highlighting a key advantage of this approach. Based on this, it follows naturally that the double robustness property, as characterized in Proposition 1, ensures the unbiasedness of our estimator.

## D    Utility of Sufficiency measure

In this section, we would like to clarify the utility of the sufficiency measure by addressing it from the following two aspects.

**Experiment Perspective**    First, for case study 1 (see Section 5.1), we select top-$K$ features with $K \in \{10, 20, 30, \text{all}\}$, calculate the sufficiency measure as well as the worst-accuracy (among $\mu_{\mathbb{P}}(\cdot)$, $\mu_{\mathbb{Q}}(\cdot)$, and $\pi(\cdot)$) of our attribution model. The results in Table 2 show a clear trend: better sufficiency (*corresponding to lower sufficiency score, which is the MSE*) correspond to improved prediction performance and more precise attribution, thereby highlighting the practical utility of this measure. *Note that lower sufficiency score represents better sufficiency, since the measure is defined as mean-square error.*

**Theory Perspective**    Second, as demonstrated in Proposition 2, the *attribution estimation error* is derived as:

$$\mathcal{O}_P\bigg(\underbrace{\max_{\tau \in \{\mathbb{P}, \mathbb{Q}\}} \mathbb{E}\Big[(\hat{\mu}_\tau(\tilde{X}_{-S}) - R_\tau(\tilde{X}_{-S}))^2\Big]^{\frac{1}{2}}}_{\text{sufficiency measure in Equation (1), error of } \mu_{\mathbb{P}}, \mu_{\mathbb{Q}}} \underbrace{\mathbb{E}\Big[(\hat{\pi}(\tilde{X}_{-S}) - \pi(\tilde{X}_{-S}))^2\Big]^{\frac{1}{2}}}_{\text{error of } \pi(\cdot)}\bigg), \qquad (15)$$

which is directly controlled by the sufficiency measure. This theoretical result further underscores the importance and relevance of this metric in ensuring robust and reliable attribution.

**Why to report the worst accuracy?**    The attribution estimation error in Equation (15) is the *product* of (1) the worst error of $\mu_{\mathbb{P}}(\cdot)$, $\mu_{\mathbb{Q}}(\cdot)$ and (2) the error of $\pi(\cdot)$. Therefore, we care about the performance of all three predictors $\mu_{\mathbb{P}}(\cdot)$, $\mu_{\mathbb{Q}}(\cdot)$ and $\pi(\cdot)$. Thus, we report the worst accuracy among $\mu_{\mathbb{P}}(\cdot)$, $\mu_{\mathbb{Q}}(\cdot)$ and $\pi(\cdot)$ to reflect the sufficiency of extracted features.

Table 2: Utility of sufficiency measure, where lower sufficiency score represents better sufficiency, since the measure is defined as mean-square error.

|  | Top-10 Features | Top-20 Features | Top-30 Features | All Features |
|---|---|---|---|---|
| Sufficiency↓ | 0.202 | 0.182 | 0.176 | 0.166 |
| Worst-Acc↑ (among $\mu_{\mathbb{P}}, \mu_{\mathbb{Q}}, \pi$) | 68.9 | 72.8 | 73.4 | 76.0 |

## E    Value of our Attribution on Interventions

In this section, we would like to demonstrate how our attribution can guide further interventions.

**What can TSSA provide**    Our attribution results are designed to provide a comprehensive framework to understand the distribution shifts responsible for performance drops. Specifically, the interpretability of both static features and extracted non-stationary properties allows clinicians to identify the underlying reasons behind model failures.

**How to leverage**    Our results provide a basis for both model-centric and data-centric intervention, and can also inform a smart deployment.

- **Model Intervention**: When the model can be finetuned/re-trained, our TSSA can guide target model interventions. For example, our results can be incorporated with distributionally robust optimization to form some directed/targted uncertainty set, as shown in Liu et al. (2023a). Also, our results can guide a more data-driven algorithmic design for better robustness or fairness among demographic groups (e.g., defined by the identified feature "Age").
- **Data-Centric Intervention**: Our results can guide efficient data collection (for example, collect more data from the risk region), data balancing (as done in our case study 2 in Section 5.2), and targeted data augmentation (only perturb sensitive features).
- **Smart Deployment**: When the model cannot be changed, our safe/risk regions can guide engineers/clinicians about where to (not to) deploy the model, as demonstrated in our case study 1 in Section 5.1 and Appendix G.3.

**How to choose**    In practice, these three aspects should be incorporated together so as to further improve the model. And we acknowledge that further efforts can be made on how to select better algorithms in each kind based on the specific situations one face, which to the best of our knowledge is still an untouched field.

## F    DETAILS OF TEMPORAL PROPERTIES

In this section, we first clarify the guidance and transparency in our metric selection, and then introduce the metrics we used in detail.

### F.1    GUIDANCE AND TRANSPARENCY IN METRIC SELECTION

Before introducing metrics used for extracting non-stationarity, we would like to clarify the guidance and transparency in our metric selection.

**Desiderata for metrics**    : To provide a principled and transparent basis for metric selection, we introduce desiderata for the types of metrics (Section 3.1). Specifically, our desiderata define the properties that metrics should capture in time-series data, grouped into four categories:

- **Global Characteristics**: Capturing long-term trends, averages, and periodicities.
- **Local Dynamics**: Measuring short-term variability and anomalies.
- **Structural Changes**: Identifying abrupt shifts in data generation processes.
- **Multivariate (Inter-Series) Relationships**: Quantifying dependencies between multiple time series.

By organizing metrics into these categories, we offer a structured framework to guide practitioners in selecting or tailoring metrics for their specific datasets. To systematically assess the quality and relevance of metrics, we employ a sufficiency measure, which quantifies the predictive power of selected metrics in explaining performance degradation. As theoretically demonstrated by Proposition 2, metrics with better sufficiency (lower sufficiency score, experiments see Appendix D) contribute enable more reliable attribution. We believe such a measure provides a data-driven measure to evaluate and refine metric selection in addition.

While we acknowledge that our current set of metrics is not exhaustive, TSSA as a framework permits users to extend it with additional metrics that align with the defined desiderata. By linking sufficiency to predictive performance, this then ensures that newly added metrics have utility for attribution.

### F.2 PROPERTY DEFINITION

To address the non-stationarity of time series, inspired from various fields, such as finance, statistics, and signal processing, we identify numerous data property metrics. Specifically, for a sequence $V_{1\ldots t}$ of length $t$, we define metrics corresponding to different temporal properties:

*1. Global Characteristics*:

- Overall Statistics: We calculate the average, standard deviation, maximum, and minimum values of each time-series feature.

- Standardized Slope: Widely used in financial and climate analysis, the standardized slope is defined as:

$$\text{Standardized Slope} := \text{Slope}(V_{1\ldots t})/\text{Std}(V_{1\ldots t}), \tag{16}$$

  which quantifies the strength of a sequence relative to its variability.

- Smoothed Trend: Drawing inspiration from analytical chemistry, we use the Savitzky-Golay filter (Savitzky & Golay, 1964) to characterize the smoothed trend. This filter smooths the data by fitting a polynomial to each segment of data points, effectively extracting the underlying trend.

- Frequency: We calculate the dominant frequency for each time series using the Fast Fourier Transform (FFT) i.e. we extract the dominant frequency (with the maximal amplitude) from positive frequencies, where the positive frequency values are real numbers, calculated as $f_k = \frac{k}{NT_s}$ ($T_S = 1$ in our experiments).

- Signal-to-Noise Ratio: We compute the Signal-to-Noise Ratio (SNR) for each feature in a time-series dataset by using a moving average to estimate the signal and residuals as noise. The metric is defined as:

$$\text{SNR} = \frac{\mathbb{E}[\|\tilde{V}\|^2]}{\mathbb{E}[\|V - \tilde{V}\|^2]}, \tag{17}$$

  where $V$ represents the original time-series feature, and $\tilde{V}$ is the smoothed data (estimated signal).

*2. Local Dynamics*:

- Breakout points: Inspired by the Bollinger Bands (Bollinger, 1992) widely applied in financial analysis, we calculate the number of breakout points within the sequence $V_{1\ldots t}$ as:

$$\left|\mathcal{V}(V_{1\ldots t})\right| := \left|\left\{i : |V_i| \geq |\text{Mean}(V_{1\ldots t})| + 2 \cdot \text{Std}(V_{1\ldots t}) \text{ for } i = 1, \ldots, t\right\}\right| \tag{18}$$

  which identifies the number of points that fall outside the 2-standard-deviation bands to capture its local non-stationarity.

- Short-term Variability: We first calculate first differences $\Delta V_i = V_i - V_{i-1}$, and the short-term variability can be defined as:

$$\sigma_{\Delta V} = \text{Std}(\Delta V_2, \ldots, \Delta V_t), \tag{19}$$

  which is the standard deviation of first differences, and a larger standard deviation indicates greater short-term fluctuation.

- High-Frequency Energy: To capture the high-frequency components, based on Discrete Fourier Transform, we define the high-frequency energy as:

$$E_{\text{high}} := \sum_{k=\lceil \frac{t}{2} \rceil}^{t-1} \sum_{j=1}^{t} V_j e^{-i2\pi kj/t}, \tag{20}$$

  which calculates the squared magnitudes of the upper half of the frequency spectrum.

- Normalized Jitter Index: To provide a comprehensive characteristic of variability, we design the normalized Jitter index as:

$$\text{Jitter Index} := \frac{\alpha \sigma_{\Delta V} + (1 - \alpha) E_{\text{high}}}{\text{mean}(|V_{1\ldots t}|)}, \tag{21}$$

  where $\alpha \in (0, 1)$ is the hyper-parameter to adjust the information from time and frequency domains to provide a comprehensive measure of fluctuation in a time series.

- Relative Strength Index: In order to capture the speed and change of a signal, we use the relative strength index (RSI) defined as:

$$\text{RS} := \frac{\sum_{i=2}^{t} \mathbb{I}(V_i > V_{i-1}) \cdot (V_i - V_{i-1})}{\sum_{i=2}^{t} \mathbb{I}(V_i < V_{i-1}) \cdot (V_{i-1} - V_i)}, \quad \text{RSI} := 100 - \frac{100}{1 + \text{RS}}, \tag{22}$$

where RS captures the ratio of average gains to average declines of $V_{1\ldots t}$. Thus, the RSI measures the momentum strength of a signal, particularly the relative magnitude of recent gains versus declines.

- KPSS Non-Stationary Test: We calculate the $p$-value from the KPSS test (Kwiatkowski-Phillips-Schmidt-Shin test), which is used to assess the stationarity of a time series.

*3. Structural Changes*

- Change points: From statistics, we utilize Pruned Exact Linear Time (PELT (Killick et al., 2012)) to capture the optimal change point set for sequence $V_{1\ldots t}$, which identifies multiple change points in the sequence such that the statistical properties (e.g., mean, variance) remain consistent within each segment.

- Trend Variability: We calculate the local trend changes associated with the change points.

*4. Multivariate Interaction*

- Covariance Variability: To capture the varying relationships (w.r.t. time) among multiple time-series features, inspired by literatures on local covariance (Papadimitriou et al., 2006; Chen et al., 2010), we design a covariance variability for time-series features $\mathbf{V}_{1\ldots T} \in \mathbb{R}^{d \times T}$ as follows:

$$\text{Cov.Var}(\mathbf{V}_{1\ldots T}) := \text{Std}(\lambda_{\max}(C(t))), \tag{23}$$

where $C(t)$ denotes the local covariance matrix at time $t$.

We list the metrics in Table 1, and more details as well as some other metrics are provided in the Appendix. With these metrics, we combine the static features $U$ with the metrics of all time-series features $V_{1\ldots t}$, collectively referred to as $\tilde{X}$ in the following sections of this paper.

# G  ADDITIONAL EXPERIMENT RESULTS

In this section, we add more details of our experiments.

## G.1  DATASET DESCRIPTIONS

**Dataset Details**  Through our experiments, we use the Medical Information Mart for Intensive Care (MIMIC) (Johnson et al., 2016) dataset, which is representative of *complex real-world* medical time series. The whole dataset contains $23,100$ patients, from which **9** static demographic features—including insurance status, marital status, ethnicity, gender, age, previous admission, previous ICU stay time, admission type, and admission location—and **53** time-series health indexes, including BUN, Braden activity, Braden friction/shear, Braden mobility, Braden moisture, Braden nutrition, GCS (eye opening), GCS (motor response), GCS (verbal response), HCO3, MCH, MCHC, MCV, O2 fraction, O2 pressure, O2 saturation, PTT, RDW, anion gap, arterial line, bicarbonate, calcium, chloride, cordis/introducer, creatinine, dialysis catheter, diastolic blood pressure, glucose, heart rate, hematocrit, hemoglobin, magnesium, mean airway pressure, mean arterial blood pressure, multi lumen, norepinephrine, pCO2, pH, pO2, phenylephrine, phosphate, PICC line, platelet count, potassium, red blood cells, respiratory rate, sodium, systolic blood pressure, temperature, tidal volume, urea nitrogen, ventilator usage, and white blood cell counts. The time-series features are measured every hour, and the average length is 85.4.

In case studies $1 \sim 3$, the outcome variable is mortality, and in case study 4, the outcome variable is ventilator usage (where we exclude the mortality feature).

**Advantages of the MIMIC Dataset**  The MIMIC dataset offers several advantages, making it especially suitable for our study:

- **Real-World Complexity**: MIMIC represents real-world clinical settings, including data from a diverse set of patients across different demographics and medical conditions. This is crucial for developing models that generalize well to actual hospital environments, where variability is high and patient trajectories are complex.

- **Granularity and Richness of Time-Series Data**: The 53 time-series features in MIMIC cover a wide array of physiological signals, lab measurements, and treatments. This granularity allows for more comprehensive modeling of patient health. In comparison, many other datasets may focus on a narrower subset of features, limiting their ability to capture nuanced patient dynamics.

- **Diverse Outcome Variables**: The dataset supports the study of multiple clinical outcomes, such as mortality and ventilator usage, which are critical in the context of ICU care. This allows for different case study designs and enables us to explore various clinical scenarios and broaden the scope of potential applications.

- **Longitudinal Data with High Temporal Resolution**: MIMIC provides high-resolution, time-stamped data that tracks patients throughout their ICU stays. This temporal depth allows for the detailed study of health trajectories over time, which is essential for building predictive models that can anticipate patient outcomes based on continuous monitoring—a capability that many smaller or less detailed datasets lack.

- **Widely Used and Well-Validated**: MIMIC has been extensively validated in the research community and is a well-established benchmark dataset for various tasks in medical machine learning. Its widespread use ensures the reliability and comparability of our results with those of previous studies. Additionally, its consistent presence in peer-reviewed research provides confidence in its data quality and facilitates benchmarking

**Justification of the Temporal Shift Setting**   Then we would like to further clarify the early diagnosis setup in case study 3. This partition reflects a practical clinical scenario where early-stage predictions are critical in healthcare to identify patients at risk. Note, we are not predicting the past but rather assessing how well a model trained on later-stage data can generalize and perform on earlier data (which is temporally shifted). This is important in healthcare since while later-stage data may eventually become available, clinicians often need to make decisions based on the first 24 hours of patient data to prevent adverse outcomes or to take actions. Hence, our setup evaluates the model's ability to perform early predictions and correctly identify patients at risk, which is essential for preemptive and early care in clinical practice.

### G.2    MODEL TRAINING DETAILS

As for the original model (under evaluation), we use Transformer model (n_head:4, n_layer:3, hidden_dim:32), learning rate is $1e^{-3}$, the total epoch number is 200, batch size is 256, and the early stop is used during training (according to last 10 epoch). As for the attribution model: The model architecture is shown in Figure 2, where we use two-layer MLP with hidden size selected from $\{16,32,64,128\}$ for each part according to the validation results, learning rate $1e^{-3}$, and batch size 64.

### G.3    DIFFERENT TEMPORAL PROPERTIES MATTER FOR DIFFERENT TIME SERIES

To illustrate the necessity of incorporating various temporal properties, as discussed in Section 3.1, we compute the feature importance scores within the conditional risk predictor $\hat{\mu}_{\mathbb{P}}(\cdot)$. The importance score for each feature is determined using the gradient norm of that feature, given by:

$$\mathbb{E}_{\mathbb{P}}\big[|\partial\hat{\mu}_{\mathbb{P}}(X)/\partial\tilde{X}_j|\big],$$

where a higher score indicates that the feature plays a more significant role in predicting the error of the deployed model $f(\cdot)$. We visualize the feature importance in Figure 6. The results reveal that different temporal metrics are important for different time-series features, highlighting the intricate

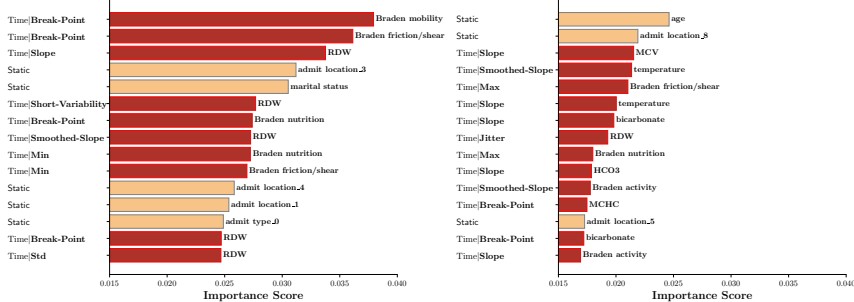

Figure 6: Feature Importance. In case study 1, we visualize the top 30 most important features for $\hat{\mu}_{\mathbb{P}}(\cdot)$. The results highlight that different temporal characteristics are significant for different time series, demonstrating the complex nature of time series.

nature of time-series data. This not only highlights the complexity of the MIMIC dataset, but also justifies the necessity of the temporal property characterization stage in our framework.

### G.4 COMPARISON WITH "STATIC" REGION ANALYSIS

In Case Study 1, a straightforward approach would be to treat all time-series features as static features and then apply the existing static region analysis method (Liu et al., 2023a). This approach offers a natural baseline for comparison, as it simplifies the temporal aspect of the data. However, this simplification may overlook important dynamic patterns inherent in the time-series data.

Table 3: Comparison of Region Analysis Methods on Test Set.

|  | Safe Region | | Risk Region | |
|---|---|---|---|---|
|  | Error Rate ↓ | # Samples ↑ | Error Rate ↑ | # Samples ↑ |
| Static | 8.8% | 215 | 36.7% | 30 |
| Ours | **3.7%** | **241** | **52.0%** | **296** |

In Table 3, we present the error rate and the number of test samples falling within each region for both the static method and our proposed method. A lower error rate indicates a better safe region, while a higher error rate corresponds to a more effective risk region. Regarding sample size, a larger number of test samples in a region suggests that the region is more robust and reliable.

From the results, it is evident that our region analysis method significantly outperforms the static feature-based approach. Additionally, the risk region in our method encompasses a much larger sample size, suggesting that our method, by incorporating temporal properties, captures more reliable and generalizable regions.

### G.5 SAMPLE EFFICIENCY OF OUR ARCHITECTURE

In case study 2, we examine the performance of our proposed architecture (Figure 2) in fitting the functions $\hat{\mu}_{\mathbb{P}}(\cdot)$, $\hat{\mu}_{\mathbb{Q}}(\cdot)$, and $\hat{\pi}(\cdot)$ across different target sample sizes. Since the outcome variables are binary, we compute the *worst balanced accuracy* of our model, using XGBoost as a baseline for comparison. For XGBoost, we train three independent models, one for each of the three functions. Note that we mainly compare with XGBoost here since it has shown superior prediction power on tabular data, and even outperforms neural networks (Gardner et al., 2022; McElfresh et al., 2024).

As shown in Figure 7, when the target sample size exceeds 30% of the training data, our proposed model consistently outperforms XGBoost, highlighting the effectiveness of the shared representation

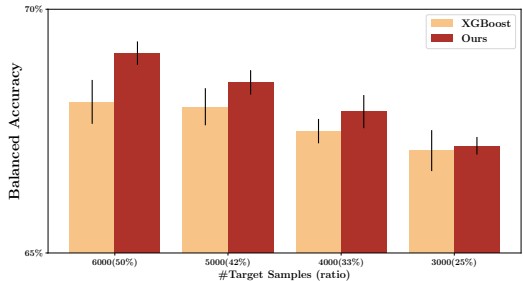

Figure 7: The worst balanced accuracy under different target sample sizes.

Table 4: Intervention results. We compare our attribution-guided simple intervention with simple finetuning.

|  | Original | Finetune | Age Intervention | Admit Loc. Intervention | ICU Stay Time Intervention |
|---|---|---|---|---|---|
| Accuracy | 73.8 | 77.0 | **83.9** | **79.2** | **84.3** |
| Macro-F1 | 63.7 | 65.7 | **68.4** | **67.0** | **69.3** |

space. However, as the target sample size decreases further, our model performs similarly to XGBoost. This outcome is expected, as tree-ensemble models like XGBoost are known to excel on tabular data, even in low-data regimes. Additionally, for our attribution objective, a wide range of models can be employed to estimate $\hat{\mu}_{\mathbb{P}}(\cdot)$, $\hat{\mu}_{\mathbb{Q}}(\cdot)$, and $\hat{\pi}(\cdot)$, offering flexibility in model selection.

### G.6 COMPARISON WITH SIMPLE FINETUNING

For case study 2, we add the results of finetuning, where we use 50% samples (3200) from the original test set for finetuning, and test on the remaining 50% test samples. For our model intervention, we mix the original training set with the finetuning samples, and then retrain a classifier with balanced weights according to the identified features (age, ICU stay time, etc.). The results are shown in Table 4, where we can see that our intervention can significantly outperform simple finetuning.

## H PROOFS

*Proof for Proposition 1.* As for unconfoundedness, the core is to prove that $R_{\mathbb{P}}(\tilde{X}_{-S})$ and $R_{\mathbb{Q}}(\tilde{X}_{-S})$ are both functions relying solely on $\tilde{X}_{-S}$. Since $\tilde{X}_{-S}$ is derived from the original input $X$ via some kind of transformations, we define $g(X) \coloneqq \tilde{X}_{-S}$. We first prove that $R_{\mathbb{P}}(\tilde{X}_{-S}) = R_{\mathbb{P}}(g(X)) = \mathbb{E}_{\mathbb{P}}[\ell(f_\theta(X), Y)|g(X)]$ is a function of $g(X)$. From measure theory, the conditional expectation $\mathbb{E}_{\mathbb{P}}[\ell(f_\theta(X), Y)|g(X)]$ is defined with respect to the $\sigma$-algebra $\sigma(g(X))$ and satisfies:

$$\mathbb{E}_{\mathbb{P}}[\ell(f_\theta(X), Y)|g(X)] = h(g(X)), \tag{24}$$

where $h(g(X))$ is a measurable function of $g(X)$. This implies that for any event $A \in \sigma(g(X))$, we have:

$$\mathbb{E}[\mathbb{E}_{\mathbb{P}}[\ell(f_\theta(X), Y)|g(X)] \cdot \mathbb{I}_A] = \mathbb{E}[\ell(f_\theta(X), Y) \cdot \mathbb{I}_A], \tag{25}$$

which implies that the conditional expectation $\mathbb{E}_{\mathbb{P}}[\ell(f_\theta(X), Y)|g(X)]$ depends only on the value of $g(X)$. Note that here we use the Tower property (Billingsley, 2017), which states that $\mathbb{E}[\mathbb{E}[X|\mathcal{G}]] = \mathbb{E}[X]$, where $\mathcal{G}$ is a sub-$\sigma$-algebra. This means that when taking the conditional expectation of a random variable $X$, and then the result is equal to the total expectation of $X$. For the above equation, the inner conditional expectation $\mathbb{E}_{\mathbb{P}}[\ell(X, Y)|g(X)]$ can be viewed as a function of $g(X)$, denoted as $h(g(X))$. And the outer expectation considers the expectation of $h(g(X))$ over some event $A \in \sigma(g(X))$. Thus, by tower property, the result of LHS equals to $\mathbb{E}[\ell(X, Y) \cdot \mathbb{I}_A]$ because $\mathbb{I}_A$ is measurable with respect to $\sigma(g(X))$, the $\sigma$-algebra generated by $g(X)$. And this ensures that $\mathbb{E}_P[\ell(X, Y)|g(X)]$ is well-defined and depends only on $g(X)$.

Based on this, we can re-write $R_\mathbb{P}(\tilde{X}_{-S})$ and $R_\mathbb{Q}(\tilde{X}_{-S})$ as $h_1(g(X))$ and $h_2(g(X))$. Then we have:

$$\Pr(T, R_\mathbb{P}(\tilde{(X)}_{-S}), R_\mathbb{P}(\tilde{(X)}_{-S})|\tilde{X}_{-S}) \tag{26}$$

$$=\Pr(T, h_1(g(X)), h_2(g(X))|g(X)) \tag{27}$$

$$=\Pr(T|h_1(g(X)), h_2(g(X)), g(X)) \cdot \Pr(h_1(g(X)), h_2(g(X))|g(X)) \tag{28}$$

$$=\Pr(T|g(X)) \cdot \Pr(h_1(g(X)), h_2(g(X))|g(X)), \tag{29}$$

which proves the conditional independence. Note that the last equation holds because $h_1(g(X))$ and $h_2(g(X))$ are both functions of $g(X)$.

Then we proceed to proving the unbiasedness (also known as the double robustness). Note that the SUTVA assumption holds naturally in our problem setting. First, if $\hat{\mu}_\mathbb{P}(\cdot)$ and $\hat{\mu}_\mathbb{Q}(\cdot)$ are consistent, i.e. $\hat{\mu}_\mathbb{P}(\cdot) = R_\mathbb{P}(\cdot)$ and $\hat{\mu}_\mathbb{Q}(\cdot) = R_\mathbb{Q}(\cdot)$, it is easy to show the consistency of our estimator (by plugging-in $\hat{\mu}_\mathbb{P}$ and $\hat{\mu}_\mathbb{Q}$). Second, if $\hat{\pi}(\cdot)$ is consistent, i.e. $\hat{\pi}(\cdot) \approx \pi(\cdot)$, denote the indicator

$$T = \begin{cases} 1, & \text{if } \tilde{X}_{-S} \text{ is from } \mathbb{Q} \\ 0, & \text{if } \tilde{X}_{-S} \text{ is from } \mathbb{P}, \end{cases} \tag{30}$$

our estimator can be re-written as:

$$\widehat{\text{Attr.}}(S) = \underbrace{\frac{1}{n_P + n_Q} \sum_{i=1}^{n_Q+n_P} \left( \frac{T_i R_\mathbb{Q}(\tilde{X}^i_{-S})}{\pi(\tilde{X}^i_{-S})} - \frac{(1-T_i)R_\mathbb{P}(\tilde{X}^i_{-S})}{\pi(\tilde{X}^i_{-S})} \right)}_{\text{optimal IPW estimator}}$$
$$+ \underbrace{\frac{1}{n_P + n_Q} \sum_{i=1}^{n_Q+n_P} \left( \hat{\mu}_\mathbb{Q}(\tilde{X}^i_{-S})(1 - \frac{T_i}{\pi(\tilde{X}^i_{-S})}) - \hat{\mu}_\mathbb{P}(\tilde{X}^i_{-S})(1 - \frac{1-T_i}{1-\pi(\tilde{X}^i_{-S})}) \right)}_{\text{additional term}}. \tag{31}$$

To complete the proof, we need to show (i) the optimal IPW estimator is consistent, and (ii) the additional term is equal to 0. The proof of consistency of the optimal IPW estimator is standard and one can refer to Wager (2020, Chapter 2). For the additional term, we have:

$$\mathbb{E}[1 - \frac{T_i}{\pi(\tilde{X}^i_{-S})}|\tilde{X}^i_{-S}] = 0, \tag{32}$$

and therefore complete the proof of unbiasedness. $\qquad\square$

*Proof of Proposition 2.* Our proof builds on the established techniques presented in (Wager, 2020, Chapter 3), with tailored adaptations and simplifications specific to our problem setting.

First, since in our problem setting, $\mu_\mathbb{P}(\cdot) = R_\mathbb{P}(c)$ and $\mu_\mathbb{Q}(\cdot) = R_\mathbb{Q}(c)$, the oracle estimator is simplified to:

$$\widehat{\text{Attr.}}^\star(S) = \frac{1}{n_P + n_Q} \sum_{i=1}^{n_P+n_Q} \left( R_\mathbb{Q}(\tilde{X}^i_{-S}) - R_\mathbb{P}(\tilde{X}^i_{-S}) \right), \tag{33}$$

where we do not have the propensity score term. Then we decompose $\widehat{\text{Attr.}}(S) - \widehat{\text{Attr.}}^\star(S)$ as:

$$\widehat{\text{Attr.}}(S) - \widehat{\text{Attr.}}^\star(S) = \frac{1}{n_P + n_Q} \sum_{i=1}^{n_P+n_Q} \Big( \hat{\mu}_\mathbb{Q}(\tilde{X}^i_{-S}) - \hat{\mu}_\mathbb{P}(\tilde{X}^i_{-S}) - R_\mathbb{Q}(\tilde{X}^i_{-S}) + R_\mathbb{P}(\tilde{X}^i_{-S}) \tag{34}$$

$$+ \frac{R_\mathbb{Q}(\tilde{X}^i_{-S}) - \hat{\mu}_\mathbb{Q}(\tilde{X}^i_{-S})}{\hat{\pi}(\tilde{X}^i_{-S})} T_i - \frac{R_\mathbb{P}(\tilde{X}^i_{-S}) - \hat{\mu}_\mathbb{P}(\tilde{X}^i_{-S})}{1 - \hat{\pi}(\tilde{X}^i_{-S})}(1 - T_i) \Big) \tag{35}$$

$$= \Delta_{\mu_\mathbb{Q}} - \Delta_{\mu_\mathbb{P}}, \tag{36}$$

where we define

$$\Delta_{\mu_\mathbb{Q}} := \frac{1}{n_P + n_Q} \sum_{i=1}^{n_P+n_Q} \left( \hat{\mu}_\mathbb{Q}(\tilde{X}^i_{-S}) - R_\mathbb{Q}(\tilde{X}^i_{-S}) + \frac{R_\mathbb{Q}(\tilde{X}^i_{-S}) - \hat{\mu}_\mathbb{Q}(\tilde{X}^i_{-S})}{\hat{\pi}(\tilde{X}^i_{-S})} T_i \right), \tag{37}$$

and define $\Delta_{\mu_{\mathbb{P}}}$ analogously. In order to prove Proposition 2, it suffices to show that $\Delta_{\mu_{\mathbb{Q}}}$ satisfies that conclusion. To prove this, we decompose $\Delta_{\mu_{\mathbb{Q}}}$ as follows:

$$\Delta_{\mu_{\mathbb{Q}}} = \frac{1}{n_P + n_Q} \sum_{i=1}^{n_P + n_Q} (\hat{\mu}_{\mathbb{Q}}(\tilde{X}^i_{-S}) - R_{\mathbb{Q}}(\tilde{X}^i_{-S}))(1 - \frac{T_i}{\hat{\pi}(\tilde{X}^i_{-S})}) \tag{38}$$

$$= \frac{1}{n_P + n_Q} \sum_{i=1}^{n_P + n_Q} (\hat{\mu}_{\mathbb{Q}}(\tilde{X}^i_{-S}) - R_{\mathbb{Q}}(\tilde{X}^i_{-S}))(1 - \frac{T_i}{\pi(\tilde{X}^i_{-S})}) \tag{39}$$

$$- \frac{1}{n_P + n_Q} \sum_{i=1}^{n_P + n_Q} T_i(\hat{\mu}_{\mathbb{Q}}(\tilde{X}^i_{-S}) - R_{\mathbb{Q}}(\tilde{X}^i_{-S}))(\frac{1}{\hat{\pi}(\tilde{X}^i_{-S})} - \frac{1}{\pi(\tilde{X}^i_{-S})}). \tag{40}$$

We first deal with the first term. Note that in practice, we typically use cross-fitting and therefore $\hat{\mu}_{\mathbb{Q}}(\cdot)$ can be viewed as deterministic in the following. From Equation (32), the summands used to build the first term become mean-zero. Therefore, we have:

$$\mathbb{E}\left[\left(\frac{1}{n_P + n_Q} \sum_{i=1}^{n_P + n_Q} (\hat{\mu}_{\mathbb{Q}}(\tilde{X}^i_{-S}) - R_{\mathbb{Q}}(\tilde{X}^i_{-S}))(1 - \frac{T_i}{\pi(\tilde{X}^i_{-S})})\right)^2\right] \tag{41}$$

$$= \text{Var}\left(\frac{1}{n_P + n_Q} \sum_{i=1}^{n_P + n_Q} (\hat{\mu}_{\mathbb{Q}}(\tilde{X}^i_{-S}) - R_{\mathbb{Q}}(\tilde{X}^i_{-S}))(1 - \frac{T_i}{\pi(\tilde{X}^i_{-S})})\right) \tag{42}$$

$$= \frac{1}{n_P + n_Q} \text{Var}\left((\hat{\mu}_{\mathbb{Q}}(\tilde{X}^i_{-S}) - R_{\mathbb{Q}}(\tilde{X}^i_{-S}))(1 - \frac{T_i}{\pi(\tilde{X}^i_{-S})})\right) \quad \text{(independent terms)} \tag{43}$$

$$= \frac{1}{n_P + n_Q} \mathbb{E}\left[(\hat{\mu}_{\mathbb{Q}}(\tilde{X}^i_{-S}) - R_{\mathbb{Q}}(\tilde{X}^i_{-S}))^2(1 - \frac{T_i}{\pi(\tilde{X}^i_{-S})})^2\right] \tag{44}$$

$$= \frac{1}{n_P + n_Q} \mathbb{E}\left[(\hat{\mu}_{\mathbb{Q}}(\tilde{X}^i_{-S}) - R_{\mathbb{Q}}(\tilde{X}^i_{-S}))^2(\frac{1}{\pi(\tilde{X}^i_{-S})} - 1)\right], \tag{45}$$

where the last equality is because of:

$$\mathbb{E}[(1 - \frac{T_i}{\pi(\tilde{X}^i_{-S})})^2 | \tilde{X}^i_{-S}] = \mathbb{E}[1 - \frac{2T_i}{\pi(\tilde{X}^i_{-S})} + \frac{T_i}{\pi^2(\tilde{X}^i_{-S})} | \tilde{X}^i_{-S}] = (\frac{1}{\pi(\tilde{X}^i_{-S})} - 1). \tag{46}$$

Then from the overlap assumption, we assume that for all $\tilde{X}^i_{-S}$, $\eta < \pi(\tilde{(X)}^i_{-S}) < 1 - \eta$, which gives that

$$\frac{1}{n_P + n_Q} \mathbb{E}\left[(\hat{\mu}_{\mathbb{Q}}(\tilde{X}^i_{-S}) - R_{\mathbb{Q}}(\tilde{X}^i_{-S}))^2(\frac{1}{\pi(\tilde{X}^i_{-S})} - 1)\right] \tag{47}$$

$$\leq \frac{1}{\eta(n_P + n_Q)} \mathbb{E}[(\hat{\mu}_{\mathbb{Q}}(\tilde{X}^i_{-S}) - R_{\mathbb{Q}}(\tilde{X}^i_{-S}))^2]. \tag{48}$$

Then for the second term, we have:

$$\frac{1}{n_P + n_Q} \sum_{i=1}^{n_P + n_Q} T_i(\hat{\mu}_{\mathbb{Q}}(\tilde{X}^i_{-S}) - R_{\mathbb{Q}}(\tilde{X}^i_{-S}))(\frac{1}{\hat{\pi}(\tilde{X}^i_{-S})} - \frac{1}{\pi(\tilde{X}^i_{-S})}) \tag{49}$$

$$\leq \sqrt{\frac{1}{n_P + n_Q} \sum_{i:T_i=1} (\hat{\mu}_{\mathbb{Q}}(\tilde{X}^i_{-S}) - R_{\mathbb{Q}}(\tilde{X}^i_{-S}))^2} \cdot \sqrt{\frac{1}{n_P + n_Q} \sum_{i:T_i=1} (\frac{1}{\hat{\pi}(\tilde{X}^i_{-S})} - \frac{1}{\pi(\tilde{X}^i_{-S})})^2} \tag{50}$$

$$= \mathcal{O}_P\left(\mathbb{E}\left[(\hat{\mu}_{\mathbb{Q}}(\tilde{X}_{-S}) - R_\tau(\tilde{X}_{-S}))^2\right]^{\frac{1}{2}} \mathbb{E}\left[(\hat{\pi}(\tilde{X}_{-S}) - \pi(\tilde{X}_{-S}))^2\right]^{\frac{1}{2}}\right). \tag{51}$$

$\square$

