# OpenReview forum: "Going Beyond Static: Understanding Shifts with Time-Series Attribution"
_ICLR.cc/2025/Conference — ICLR 2025 Poster_

### Official Review · Reviewer_WKbg · 2024-10-29

**Soundness:** 3
**Presentation:** 2
**Contribution:** 2
**Rating:** 6
**Confidence:** 3

**Summary:**

The paper presents methods for understanding and attributing predictive performance drops in time series classification models due to temporal data properties.

**Strengths:**

- The paper gathers and systematizes metrics to attribute performance shifts in time series.
- The paper present four different case studies on clinical scenarios with distribution shifts.
- The paper uses real-world healthcare data with a large patient population.
- The paper discusses methods to identify the specific temporal factors driving prediction error, providing interpretability and guidance for targeted domain adaptation methods.

**Weaknesses:**

1. Clarity Regarding Prior Work: This paper introduces a Sufficiency Measure and a method for performance drop attribution, claiming to be the first to analyze detailed shift patterns in time-series data. However, distribution shifts in time series are not a new topic. A clearer discussion of prior work, specifically addressing how this study differs from existing approaches, would better position and motivate the contribution.

2. Novelty Concerns: The novelty of the methods appears limited. For instance, neither the metrics predicting classification error nor the models used are new. The primary innovation seems to lie in the combined use of these metrics to predict and understand performance drops (i.e., using them as inputs to a model that estimates the prediction error of another model).

3. Utility of the Sufficiency Measure: The Sufficiency Measure is introduced to assess the optimal predictive power of the metrics, with empirical studies reporting values of around 0.16 for mean squared error and 0.25 for mean balanced 0-1 loss. This information would be better suited to the results or case study sections. Additionally, it is unclear what constitutes a "good" sufficiency measure. Experiments comparing various sufficiency measures and their ability to identify ‘safe’ and ‘risk’ regions (Case Study 1) would help motivate this proposed measure.

4. Clarity in Case Studys:
- The case studies lack clear explanations of key methodological details, such as the transformer model architecture, model, hyperparameters, and evaluation metrics. Including summary paragraphs in the Methods section for the patient cohort, models/model training, and evaluation used across all four case studies would enhance clarity.
- Case Study 3: The data partition strategy is unusual, with the validation set (the last 24 hours) occurring temporally after the test set (the first 24 hours). The practicality of this experiment is unclear, given the unrealistic and artificial temporal shift.

5. Utility of the Approach/Lack of Baselines:
- The lack of baseline comparisons or ablations studies makes it challenging to evaluate the contributions of the proposed methods.
- The work advocates for identifying specific temporal factors that contribute to performance decline, as opposed to solely applying robust methods or fine-tuning on new test data. However, the results do not compare this approach with other common methods, such as fine-tuning, used to adapt data to new distributions. Including such comparisons would strengthen the motivation for this approach, particularly by illustrating potential computational benefits and performance trade-offs relative to fine-tuning.
- Additionally, the interpretability of performance drop attribution appears valuable for guiding data reweighting or balancing techniques for certain features (e.g., demographic features in Case Study 2). However, it remains unclear what interventions would be applied to address these temporal factors or whether these identified factors would be deemed clinically useful by experts.

**Questions:**

Questions:
1. Case Study 2: The authors mention reweighting the data based solely on the inverse density ratio. Which dataset is this applied to?
2. The paper mentions prior work on performance drop attribution using Shapley values designed for static data (line 127). However, if the temporal properties discussed include summary statistics of a time series, such as mean, variance, or trend, could Shapley values then be used?

Suggestions:
- Clinical expert feedback on the relevance and actionability of identified temporal data properties would help motivate the proposed approach in practice.
- Including distribution plots of the training and test set features could help illustrate the distribution shifts for readers.

---

> ### Author Response · Authors · 2024-11-18
>
> Thank you so much for your advice to improve our paper. We have addressed your concerns as follows, and we will incorporate these into our final version.
>
> ### **Q1. Discussion with previous works about shift in time-series data**
> There’s been lots of prior works on detecting shifts in time-series data, and we discuss them in Appendix E. In time-series anomaly detection, they focus primarily on detecting breakpoints. However,
> - Our goal is to understand why a model’s performance declines in a prediction task by attributing this drop to specific properties of interest.
> - The shifts considered in time-series anomaly detection are typically associated with breakpoints, whereas our work covers a broader range of temporal properties
> - Our method can also be used to diagnose performance drops in anomaly detection methods as well.
>
> Thanks for your advice, and we will move the related work into the main body to improve the clarity.
>
> ### **Q2. Novelty**
> We would like to clarify that our contributions are significant at the problem-solving level, methodology level, and theory level:
> - **Setting level**: We focus on time-series settings, where we mainly consider the shift in non-stationary properties. This setting is ***much more complicated***, and we are the first to deal with this. Previous methods cannot deal with this effectively. As shown in Appendix D.3 (Table 2), static region analysis cannot produce reliable safe/risk regions.
> - **Methodology level**: Our attribution method is ***novel and different from previous ones***. We formulate the attribution problem as an average treatment effect estimation problem, marking, to the best of our knowledge, the first instance of this approach. And we derive the doubly robust estimator to estimate the attribution score.
> - **Theory level**: Our attribution has a solid theoretical foundation. Rather than simply incorporate average treatment effect estimation, we ***rigorously prove the unconfoundedness assumption within our framework—a substantial advancement over traditional causal inference literature***, where this assumption is typically taken for granted. Based on this, we prove the double robustness of our estimator. Our framework thus paves the way for a novel perspective on performance attribution, creating opportunities for further exploration in this domain.
>
> ### **Q3. Utility of Sufficiency measure**
>  Thank you for your valuable suggestion. We would like to clarify the utility of the sufficiency measure by addressing it from two aspects:
> - Experiment: we select top-K features with $K \in \\{10,20,30,all\\}$, calculate the sufficiency measure as well as the worst-accuracy (among $\mu_{\mathbb P}$, $\mu_{\mathbb Q}$, and $\pi$) of our attribution model. The results show a clear trend: higher sufficiency scores correspond to improved prediction performance and more precise attribution, thereby highlighting the practical utility of this measure.
>
> |          | Top-10 Features |Top-20 Features | Top-30 Features | All Features |
> |:--------:|:--------:|:--------:|:----------------:|:-----------------------:|
> | Sufficiency $\downarrow$ |  0.202 | 0.182 | 0.176| 0.166   |
> | Worst-Acc $\uparrow$ |   68.9   |   72.8   |       73.4       |           76.0          |
>
> - **Theory**: As demonstrated in Proposition (2), the attribution estimation error is directly controlled by the sufficiency measure. This theoretical result further underscores the importance and relevance of this metric in ensuring robust and reliable attribution.
>
> ### **Q4. Clarity in case studies**
> Thank you for your suggestions. We add more details and demonstrations as follows:
> - Experimental details:
>     - Original model (under evaluation) architecture: we use Transformer model (n_head:4, n_layer:3, hidden_dim:32), learning rate is $1e^{-3}$, the total epoch number is 200, batch size is 256, and the early stop is used during training (according to last 10 epoch);
>     - Attribution model: The model architecture is shown in Figure 2, where we use two-layer MLP with hidden size 32, learning rate $1e^{-3}$, and batch size 64.
> - Splitting criterion in Case Study 3: The setting is to simulate the early diagnosis [1], where we hope the model can predict the outcome as early as possible. This problem is important in real-world health-care applications.
>
> We will add these into the Appendix for better clarity.
>
> [1] Wang, Lulu. "Early diagnosis of breast cancer." Sensors 17.7 (2017): 1572.
>
> [2] Yuan, Kuo-Ching, et al. "The development an artificial intelligence algorithm for early sepsis diagnosis in the intensive care unit." International journal of medical informatics 141 (2020): 104176.

---

> > ### Author Response · Authors · 2024-11-18
> >
> > ### **Q5. Baselines**
> > Thank you for your suggestions. We ***have (in Appendix) and add more baselines*** to further validate our effectiveness:
> > * **For safe/risk region analysis**: In Appendix D.3, we compare our methods with “static” methods to demonstrate the need to consider non-stationary properties.
> > * **For sample-efficiency or sufficiency measure**: In Appendix D.4, we compare our model with XGB (SOTA for tabular prediction and does not share representations) to show that our shared representation can indeed improve the performance (with a better sample complexity).
> > * **For model intervention in Case Study 2**: We add the results of finetuning, where we use 50% samples (3200) from the original test set for finetuning, and test on the remaining 50% test samples. For our model intervention, we mix the original training set with the finetuning samples, and then retrain a classifier with balanced weights according to the identified features (age, ICU stay time, etc.). The results are as follows, where we can see that our intervention can significantly outperform simple finetuning.
> >
> > |          | Original | Finetune | Age Intervention | Admit Loc. Intervention | ICU Stay Time Intervention |
> > |:--------:|:--------:|:--------:|:----------------:|:-----------------------:|:--------------------------:|
> > | Accuracy |   73.8   |   77.0   |       83.9       |           79.2          |            84.3            |
> > | Macro-F1 |   63.7   |   65.7   |       68.4       |           67.0          |            69.3            |
> >
> > According to your suggestions, we will move these results into the main body.
> >
> >
> > ### **Q6. How to choose the interventions**
> > Thank you for raising this thoughtful question. We deeply appreciate your consideration of the practical implications of our work.
> > * **What can TSSA provide**: Our attribution results are designed to provide a comprehensive framework to understand the distribution shifts responsible for performance drops. Specifically, the interpretability of both static features and extracted non-stationary properties allows clinicians to identify the underlying reasons behind model failures.
> >
> > * **How to leverage**: Our results provide a basis for both model-centric and data-centric intervention, and can also inform a smart deployment.
> >      * **Model intervention**: When the model can be finetuned/re-trained, our TSSA can guide target model interventions.
> >      * **Data-centric intervention**: Our results can guide efficient data collection (for example, collect more data from the risk region), data balancing (as done in our intervention), and targeted data augmentation (only perturb sensitive features)
> >      * **Smart deployment**: When the model cannot be changed, our safe/risk regions can guide engineers/clinicians about where to (not to) deploy the model
> > * **How to choose**: In practice, these three aspects should be incorporated together so as to further improve the model. And we acknowledge that further efforts can be made on how to select better algorithms in each kind based on the specific situations one face, which to the best of our knowledge is still an untouched field.
> >
> > ### **Q7. Case study 2: what dataset**
> > We use MIMIC-III in case study 2. Dataset details are introduced in Appendix D.1. As for preprocessing (like missing value imputation), we follow the steps in [1].
> >
> > [1] Jarrett, Daniel, et al. "Clairvoyance: A Pipeline Toolkit for Medical Time Series." International Conference on Learning Representations.
> >
> > ### **Q8. Can we use shapley values**
> > - **Why we don’t use Shapley value**: As discussed in Appendix C, we focus on the impact of one single data property on the model’s performance degradation, while keeping all other properties (or features) constant. As a result, a full calculation of Shapley values may be unnecessary and less suitable for our objectives. Furthermore, the calculation of Shapley value is not cheap.
> > - **Our method is compatible with Shapley value**: Our proposed attribution can be viewed as the value function in Shapley value, which means that it is compatible with Shapley value.

---

> > > ### Author Response · Authors · 2024-11-22
> > >
> > > Dear Reviewer WKbg
> > >
> > > We are sincerely grateful for your time and efforts in the review process.
> > >
> > > We hope that our responses have been helpful. Given the limited time left in the discussion period, please let us know if there are any leftover concerns and if there is anything else we could do to address any further questions or comments. We are looking forward to your further feedback!
> > >
> > > Thank you!
> > >
> > > Paper Authors

---

> > > > ### Comment · Reviewer_WKbg · 2024-11-23
> > > >
> > > > Thank you for addressing my questions and concerns in detail. However, I still have some concerns regarding your response.
> > > >
> > > > Q.1: The authors mentioned, “we will move the related work into the main body to improve the clarity.” However, I do not see the related work section in the appendix. Was it moved to the main body in an updated draft? If so, marking changes in a different color would be helpful to clearly identify the updates.
> > > >
> > > > Q.2:
> > > > - Thank you for clarifying the contributions. I understand that this work addresses shifts in non-stationary properties. However, my primary concern lies with the novelty of the methods used to do so. Specifically, the non-stationary properties appear to be encoded as static features using summary statistics (e.g., mean, max, slope), which are then used as features to predict the model's prediction error. Additionally, the authors mention that they formulate the attribution problem as an average treatment effect (ATE) estimation problem. My understanding of this framework is that it typically focuses on determining the causal effect of a treatment or intervention on an outcome. Patient vital time series (e.g., heart rate, blood pressure) can reflect treatment effects, but this approach seems to treat aspects of the vitals themselves as “treatments” to predict outcomes that could be defined based on these same vitals (e.g., heart rate < 0 bpm → mortality). Treating vitals as "treatments" conflates causality, which could violate key assumptions of causal inference. Given this, I believe the ATE estimation framework with the selected healthcare time series requires more discussion in the paper for clarity.
> > > > - Regarding Setting Level D.3, this seems more like an ablation study on target sample size rather than addressing the reviewer’s request for an ablation on methods that handle shifts in non-stationary properties. Additionally, I was unable to locate Table 2 in the paper—please clarify if I missed it.
> > > >
> > > > Q.3: I appreciate the addition of the sufficiency measure ablation study. The authors state, “The results show a clear trend: higher sufficiency scores correspond to improved prediction performance and more precise attribution.” From the provided table, I see that higher sufficiency scores are associated with lower worst-accuracy measures. However, this claim could benefit from clarification. Additionally, “worst-accuracy” is not a commonly used metric, and I suggest explaining why “worst-accuracy” was selected or also comparing it to more traditional metrics, such as standard accuracy.
> > > >
> > > > Q.4: I still don't understand how the data partition—where the training set (the last 24 hours) temporally follows the test set (the first 24 hours)—is valid or useful for "simulating early diagnoses." Having a model is trying to predict a state that precedes the information it learned is not typical.
> > > >
> > > > Q.5: Thank you for clarifying the baseline for two of the experiments and the new fine-tuning experiment. Incorporating these into the paper would help demonstrate the utility of your approach.
> > > >
> > > > Q.6: I think these points are excellent contributions of the method. I would recommend framing experiments to focus and demonstrate each point and test rigorously against more baselines.
> > > >
> > > > Q.7: Thank you for clarifying the dataset and dataset details in the appendix.
> > > >
> > > > Q.8: To validate the benefits of this approach compared to similar methods, computing Shapley values for a few features and comparing them with those selected by your method would be insightful. Alternatively, a computational tradeoff analysis could highlight the potential efficiency of your approach. You could also present the compatibility statement mathematically in the paper.
> > > >
> > > > Based on the points above, I still have concerns regarding the structure, soundness, and novelty of the work, which I believe require another round of revision. While I see the model training details included in red text in the updated draft, the paper does not seem to reflect the other proposed updates, such as the new experiments or clarified related work.

---

> > > > > ### Author Response · Authors · 2024-11-24
> > > > >
> > > > > Thank you so much for your response! Based on your suggestions, **we have improved the paper** and include the following parts:
> > > > >
> > > > > ### **1. Paper Improvements**
> > > > > All revisions are highlighted in *red*.
> > > > > - **Q.1**  Related Work:  In ***Appendix A***, we add one page for related work, where we discuss about distribution shifts, time-series anomaly detection, and feature SHAP.
> > > > > - **Q.2**:
> > > > >    - In ***Appendix B***, we discuss the relationship with ATE, where we demonstrate that we only "interpret" our objective function as a special kind of ATE. But we would like to clarify that our TSSA is an attribution approach, which is ***not*** designed to estimate ATE or solve causal problems. (*see below for detailed explanations*)
> > > > >    - In ***Appendix G.4*** Table 3 (we add one table above so that Table 2 becomes Table 3), we compare our region analysis with static algorithms to show the need to consider non-stationary properties.
> > > > > - **Q.3** Sufficiency Measure: In ***Appendix C***, we first demonstrate the utility of sufficiency measure from theoretical and experimental aspects, and then we clarify why we report the worst accuracy. (*see below for detailed explanations*)
> > > > > - **Q.4** Data Partition for Temporal Shifts: In ***Appendix G.1***, we add one paragraph to demonstrate the rationales of the temporal shift setting. (*see below for detailed explanations*)
> > > > > - **Q.5** Baselines: In ***Section 5.2*** (Figure 4), we add the results of finetuning baseline. In ***Appendix G.4*** Table 3, we compare our region analysis with static algorithms to show the need to consider non-stationary properties.
> > > > > - **Q.6** How to leverage attribution results: In ***Appendix D***, we demonstrate in detail what our TSSA can provide, how to leverage the results, and how to choose the interventions.
> > > > > - **Q.8** Compatibility with Shapley Value: In ***Appendix E***, we provide the mathematical formulations in Equation (15) on how to integrate our TSSA into Shapley Value, which we think is a promising future direction for us to extend the work, and we leave this for future work.  (*see below for detailed explanations*)
> > > > >
> > > > > ### **Q.2. With ATE**
> > > > > - **Relationship with Average Treatment Effect** (ATE): First, we would like to clarify that our TSSA is an attribution approach, which is not designed to estimate ATE or solve causal problems. And we only "interpret" our objective function as a special kind of ATE.
> > > > > That is, denote \( T \in \{0,1\} \) as an indicator variable:
> > > > >
> > > > >   $T = 0$, if $X_{-S}$  is from  $\mathbb{P}$; $T = 1$, if $X_{-S}$  is from $\mathbb{Q}$,
> > > > >
> > > > >   which can be likened to a *treatment variable*.
> > > > >   And our attribution objective can then be rewritten as:
> > > > > 	\begin{equation}
> > > > > 		Attr.(S) = \mathbb{E}[R(X_{-S}, T=1) - R(X_{-S}, T=0)],
> > > > > 	\end{equation}
> > > > >    where $R(\cdot, T=1)$ denotes $R_{\mathbb{Q}}(\cdot)$, and $R(\cdot, T=0)$ denotes $R_{\mathbb{P}}(\cdot)$ that can be viewed as two *outcome function* (like in causal literatures).
> > > > >
> > > > >    Therefore, when studying the effect of one feature to the performance drop, our attribution approach controls all the other attributes (to be identical between group $T=1$ and group $T=0$), and then calculate its effect on performance drop. **Note that we are not studying causal problems**.
> > > > > - **Theoretical Analysis Beyond ATE**: Unlike in causal inference literature, where verifying the unconfoundedness assumption can be challenging, our problem formulation enables us to **prove** the unconfoundedness directly in Proposition 1, highlighting a key advantage of this approach.
> > > > > Based on this, it follows naturally that the double robustness property, as characterized in Proposition 2, ensures the unbiasedness of our estimator.
> > > > >
> > > > > ### **Q.3 Sufficiency Measure**
> > > > > - As for the score, a lower score means better sufficiency, since the score is defined as the mean-square error. We've clarified this in the caption of Table 2 to avoid any misunderstandings.
> > > > > - As for the worst accuracy, since the attribution estimation error in Proposition 2 is the **product** of (1) the worst error of $\mu_{\mathbb P}(\cdot),\mu_{\mathbb Q}(\cdot)$ and (2) the error of $\pi(\cdot)$. Therefore, we care about the performance of all three predictors $\mu_{\mathbb P}(\cdot),\mu_{\mathbb Q}(\cdot)$ and $\pi(\cdot)$. Thus, we report the worst accuracy among $\mu_{\mathbb P}(\cdot),\mu_{\mathbb Q}(\cdot)$ and $\pi(\cdot)$ to better reflect the sufficiency of extracted features.}

---

> > > > > > ### Author Response · Authors · 2024-11-24
> > > > > >
> > > > > > ### **Q.4 Data Partition for Temporal Shifts**
> > > > > > We would like to further clarify the early diagnosis setup in case study 3.
> > > > > > - This partition reflects a practical clinical scenario where early-stage predictions are critical in healthcare to identify patients at risk. Note, **we are not predicting the past but rather assessing how well a model trained on later-stage data can generalize and perform on earlier data (which is temporally shifted)**, that is, all the time-series features used in testing are also from the earlier stage.
> > > > > > - This is important in healthcare since while later-stage data may eventually become available, clinicians often need to make decisions based on the first 24 hours of patient data to prevent adverse outcomes or to take actions.
> > > > > > - Hence, our setup evaluates the model’s ability to perform early predictions and correctly identify patients at risk, which is essential for preemptive and early care in clinical practice.
> > > > > >
> > > > > > ### **Q.8 Compatibility with Shapley Value**
> > > > > > Our proposed attribution method is compatible with SHAP values, as our attribution score can be integrated into SHAP as the ``effect'' function. The mathematical formulation is as follows:
> > > > > > $$
> > > > > > \widehat{\text{SHAP.Attr}.}(S) = \sum_{\mathcal V \subseteq X_{-S}}\frac{|\mathcal V|!(d-|\mathcal V|-1)!}{d!}(\widehat{\text{Attr.}}(\mathcal V \cup {S})-\widehat{\text{Attr.}}(\mathcal V)),
> > > > > > $$
> > > > > > where $d$ denotes the number of extracted features, $X_{-S}$ denotes the set of all extracted features except $S$.
> > > > > > Therefore, our TSSA approach is compatible with Shapley Value. Given its computational cost, we can't do it experimentally so quickly, and we leave as a promising way of future extension of this work.
> > > > > >
> > > > > > Furthermore, in **Appendix E**, we demonstrate in detail why we did not use Shapley Value.
> > > > > >
> > > > > > Thank you so much for your insightful suggestions, and we are looking forward to your feedback!

---

> > > > > > > ### Author Response · Authors · 2024-11-25
> > > > > > >
> > > > > > > Dear Reviewer WKbg,
> > > > > > >
> > > > > > > We would like to express our sincere gratitude for the time and effort you dedicated to reviewing our manuscript. Your insightful feedback has been invaluable in helping us enhance the quality of our work. Given the improvements made in the revised version based on your suggestions, we kindly ask you to reconsider your original score. The initial rating of 3 was based on the earlier draft, which did not fully reflect the changes we have implemented. We believe that the updated version reflecting all your suggested changes is much improved, and we hope it meets your expectations.
> > > > > > >
> > > > > > > Thank you once again for your thoughtful review and support. We truly appreciate your consideration of this request.
> > > > > > >
> > > > > > > Best regards,
> > > > > > >
> > > > > > > Paper Authors

---

> ### Comment · Reviewer_WKbg · 2024-11-25
>
> Thank you for addressing my points and incorporating the revisions in red text. I appreciate the updates and clarifications, as they have improved my understanding. I have updated my score.
>
> Regarding case scenario 3, I recognize that the purpose is to evaluate a model in a temporally shifted scenario rather than predicting the past. My concern was about the potential implications if the training, validation, and test patient cohorts overlapped. However, the paper clarifies that the training set is a separate subset of patients, which resolves this concern.
>
> I believe the paper could still benefit from greater clarity and organization, particularly with the new updates. For example, the sentences, “Additional demonstrations on the relationship between our TSSA approach and average treatment effect (ATE) estimation can be found in Section 6. And Appendix E illustrates how our approach can be integrated with Shapley Value,” appear at the end of the methods section, even though ATE and Shapley Value are not discussed earlier in that section. I recommend repositioning this information to better align with where these topics are first introduced in the paper. For instance, it may be more effective to consolidate all mentions of Shapley Value (e.g., related work, why Shapley Value is not used, and compatibility with Shapley Value) into a single section in the appendix, while referencing this section when the topic is mentioned in the related work. Similarly, when treatment infection estimation is first referenced in line 228, it would be helpful to immediately include the sentence, “Additional demonstrations on the relationship between our TSSA approach and average treatment effect (ATE) estimation can be found in Section 6.” This restructuring would improve cohesion and readability for readers.

---

> > ### Author Response · Authors · 2024-11-26
> >
> > Thank you so much for your support and valuable suggestions! We have revised the paper to improve clarity, incorporating your feedback. Specifically, we made the following changes:
> >
> > - We now refer to Appendix C, which demonstrates the relationship with the ATE, right after the treatment effect estimation is first mentioned in line 232.
> > - All mentions of the Shapley value—covering related work, why it is not used, and its compatibility with our approach—have been consolidated into a single section in Appendix B, immediately following the related work section (Appendix A). We now refer to this section at the end of Appendix A, line 788.
> > - We refer to Appendix B (the integrated Shapley Value section) after introducing our attribution approach in line 279.
> > - We refer to Appendix D (the utility of sufficiency measure section) after introducing our sufficiency measure in line 221.
> >
> > Thank you once again for your insightful feedback and careful review. We greatly appreciate your time and effort in helping us improve the paper.

---

### Official Review · Reviewer_itss · 2024-11-01

**Soundness:** 3
**Presentation:** 3
**Contribution:** 3
**Rating:** 8
**Confidence:** 2

**Summary:**

In this paper, the authors propose a novel approach, Time-Series Shift Attribution (TSSA), to address distribution shift issues in time-series data. This method is designed to analyze application-specific patterns of distribution shifts. The authors begin by outlining desirable characteristics for time-series metrics, examining factors such as Global Characteristics, Local Dynamics, Structural Changes, and Inter-series Relationships. Following this, they introduce a performance drop attribution strategy based on the Augmented IPW and prove that their estimator is unconfounded, unbiased, and consistent. The paper concludes with case studies that validate the effectiveness of the TSSA approach.

**Strengths:**

1. The topic is highly relevant to the themes of the ICLR conference.
2. The problem addressed is critical for real-world applications and holds substantial potential for future use.
3. The proposed attribution strategy is intuitive and well-suited to the problem.

**Weaknesses:**

1. **Insufficient Baseline Comparisons**: The experimental section lacks sufficient baseline comparisons, which would further validate the effectiveness of the proposed approach.
2. **Convergence Property**: In optimization, demonstrating uniqueness (and potentially consistency) is often followed by a proof of convergence properties [1]. It appears that convergence analysis is missing in this manuscript.
3. **Font Size**: Some font sizes in Figure 5 are too small, making the content difficult to read.
4. **Error Bars in Figure 7**: The presentation of Figure 7 is inconsistent with Figures 4 and 5(a), where error bars are provided. Should this discrepancy be addressed?

---

References:
[1]. { Euclidean, Metric, and Wasserstein } Gradient Flows: an overview

**Questions:**

1. Is Equation (21) derived using the tower property?
2. The frequency is involved as discrepancy metric in Section 3.1. Is it justified in the proposed framework. For example, the distribution discussed in this paper is supported on $\mathbb{R}$.

---

> ### Author Response · Authors · 2024-11-18
>
> Thank you so much for your positive review! We would like to address your concerns as follows:
>
> ### **Q1.  Baseline Comparisons**
> Thank you for your suggestions. We ***have (in Appendix) and add more baselines*** to further validate our effectiveness:
> * **For safe/risk region analysis**: In Appendix D.3, we compare our methods with “static” methods to demonstrate the need to consider non-stationary properties.
> * **For sample-efficiency or sufficiency measure**: In Appendix D.4, we compare our model with XGB (SOTA for tabular prediction and does not share representations) to show that our shared representation can indeed improve the performance (with a better sample complexity).
> * **For model intervention in Case Study 2**: We add the results of finetuning, where we use 50% samples (3200) from the original test set for finetuning, and test on the remaining 50% test samples. For our model intervention, we mix the original training set with the finetuning samples, and then retrain a classifier with balanced weights according to the identified features (age, ICU stay time, etc.). The results are as follows, where we can see that our intervention can significantly outperform simple finetuning.
>
> |          | Original | Finetune | Age Intervention | Admit Loc. Intervention | ICU Stay Time Intervention |
> |:--------:|:--------:|:--------:|:----------------:|:-----------------------:|:--------------------------:|
> | Accuracy |   73.8   |   77.0   |       83.9       |           79.2          |            84.3            |
> | Macro-F1 |   63.7   |   65.7   |       68.4       |           67.0          |            69.3            |
>
> According to your suggestions, we will move these results into the main body.
>
> ### **Q2. Additional theoretical results**
> The convergence results align closely with traditional findings on SGD, and we do not present them as a novel contribution. Additionally, CLT results can be further expanded upon our consistency results.
>
> ### **Q3. Font size \& Error bars**
> Thank you very much for your advice! We have updated the paper (Figure 5 and Figure 7) to address these typos.
>
> ### **Q4. Equation (21)**
> Yes, it is derived from  the tower property.
>
> ### **Q5.  Frequency**
> The frequency metric used in our paper is in $\mathbb R$.

---

> > ### Author Response · Authors · 2024-11-22
> >
> > Dear Reviewer itss
> >
> > We are sincerely grateful for your time and energy in the review process. Thank you so much for your support!
> >
> > We hope that our responses have been helpful. Given the limited time left in the discussion period, please let us know if there are any leftover concerns and if there is anything else we could do to address any further questions or comments. We are looking forward to your further feedback!
> >
> > Thank you!
> >
> > Paper Authors

---

> > > ### Comment · Reviewer_itss · 2024-11-22
> > >
> > > Thank you for addressing my questions. However, the reviewer still have some concerns regarding your response and the revised manuscript:
> > >
> > > 1. In the field of signal processing, to the best of the reviewer's knowledge, the results of FFT are in the complex domain $\mathbb{C}$, rather than the real domain $\mathbb{R}$. Could the authors clarify whether the proposed framework can still be applied in this context?
> > >
> > > 2. The error bars in the authors' results should ideally represent a confidence interval, such as $1-\sigma$. Could the authors confirm this and provide more details on how the error bars were computed?
> > >
> > > 3. The reviewer noticed that the authors have included additional information regarding the hyperparameters of the neural network. Would it be necessary to conduct a sensitivity analysis to evaluate the robustness and applicability of the proposed approach under varying hyperparameter settings?
> > >
> > > 4. Regarding the derivations, if the tower property was utilized, it would be helpful to explicitly illustrate this step and cite relevant references to support its usage.
> > >
> > > Thank you for considering these points, and I look forward to your response.

---

> ### Author Response · Authors · 2024-11-22
>
> Thank you so much for your reply! We would like to address your concerns as follows:
> 1. For the frequency:
>    - The output of FFT is in $\mathbb C$, denoted as $FFT[k]=a_k+ib_k$.
>    - We then calculate the amplitudes as: $\sqrt{a_k^2+b_k^2}$.
>    - Finally we extract the **dominant frequency** (with the maximal amplitude) from positive frequencies, where the positive frequency values are **real numbers**, calculated as $f_k = \frac{k}{NT_s}$ (in our experiments we set $T_s=1$).
>
> 2. For the error bar: The error bar represents the **2 times of the standard error (i.e. $\pm$2xSE)**, calculated as $2*\frac{s}{\sqrt{n}}$, where $s=\sqrt{\frac{1}{n-1}\sum_{I=1}^n(x_i-\overline{x})^2}$ denotes the sample standard deviation, and $n$ denotes the time of repeated experiments (in our experiments $n=10$).
> 3. For the hyper-parameters:
>     - for the Transformer model under evaluation: We select a set of standard hyperparameters to obtain a transformer model, which remains fixed during evaluation. Our evaluation results are based on this model. Since our method focuses on evaluating and diagnosing (any) model's failures, it is **adaptable to various hyperparameter configurations**. Different hyperparameters can be applied, and *the evaluation results will correspond to the model configured with those settings*.
>     - for the attribution model: we choose the hidden size in $\\{16, 32, 64\\}$ and select the best parameter (32) according to the cross-validation results (accuracy of $\mu_{\mathbb P}(\cdot), \mu_{\mathbb Q}(\cdot), \pi(\cdot)$). As for the optimizer, we use Adam optimizer with the default (standard) learning rate ($1e^{-3}$). We will clarify this in the final version.
> 4. For the tower property: We add more detailed illustrations below:
>    - Tower Property:
> The tower property states that $\mathbb{E}[\mathbb{E}[X | \mathcal{G}]] = \mathbb{E}[X]$, where $\mathcal{G}$ is a sub-$\sigma$-algebra. This means that when taking the conditional expectation of a random variable $X$, and then the result is equal to the total expectation of $X$.
>    - For our equation (21), the inner conditional expectation $\mathbb{E}_{\mathbb P}[\ell(X, Y) | g(X)]$ can be viewed as a function of $g(X)$, denoted as $h(g(X))$. And the outer expectation considers the expectation of $h(g(X))$ over some event $A \in \sigma(g(X))$.
>    - Here by tower property, the result of LHS equals to $\mathbb{E}[\ell(X, Y) \cdot \mathbb{1}_A]$
>    because $\mathbb{1}_A$ is measurable with respect to $\sigma(g(X))$, the $\sigma$-algebra generated by $g(X)$. And this ensures that $\mathbb{E}_P[\ell(X, Y) | g(X)]$ is well-defined and depends only on $g(X)$.
>
> We will add this illustration into Appendix in the final version. As for the reference, we will add [1] to our proof.
>
> Thank you so much for your advice!
>
> [1] Billingsley, Patrick. Probability and measure. John Wiley & Sons, 2017.

---

### Official Review · Reviewer_36tU · 2024-11-03

**Soundness:** 4
**Presentation:** 4
**Contribution:** 3
**Rating:** 8
**Confidence:** 2

**Summary:**

This paper builds a distribution shift explainer based on a rigorous Double Robustness estimator, applied to various extracted statistics. They provide theoretical justification of the estimator formulation, and experiments show how causes of shifts can be correctly identified and how the risk regions of the original model can be assessed.

**Strengths:**

The paper is very clear, with adequate contextualization (with both reference to ML community studies and medicine-focused studies), easy to understand preliminaries and properties, precise experiment protocol and separate experiments takeaways.

The experiments are strong evidences of the method relevance, as correct domain knowledge on well known datasets is infered.

The theory is sound and easy to verify (concise, clear, and refers to the relevant reference).

**Weaknesses:**

My understanding is that no code was provided for results reproducibility.

Otherwise, I can only say that the weakness of this paper are beyond my abilities to review. The only weakness I could think of would be how much of a strong evidence the risk assessment would be to a medical practitioner. But I do not focus on the medical field, so I cannot be sure if this is an actual limitation.

**Questions:**

Wouldn't the shared representation space of the neural network compromise the double robustness by associating estimation of Pi and mu? It is likely that if one is wrong, the other is also due to their association ; hence negating the double robustness benefit. Would it be possible to test the reliability of double robustness in such conditions?

(39) assumes iid data ==> It would be good to make explicit in the main body of the paper the assumption (X,Y) iid. As the work deals with time series, I wasn't sure if repeated measures for a patient, for instance, could be used as several samples. It appears that it is not the case, and should be put in preliminaries paragraph 1.

---

> ### Author Response · Authors · 2024-11-18
>
> Thank you so much for your positive review! We would like to address your concerns as follows:
>
> ### **Q1. Code release**
> Thank you very much for your advice. The code is based on PyTorch and our algorithm is easy to implement. We promise to release our code after acceptance.
>
> ### **Q2. Double robustness network architecture**
> Thank you for your thoughtful suggestions. We would like to demonstrate more on the double robustness and our model architecture:
> * **Double robustness**: Our attribution approach is doubly robust, meaning that as long as either the outcome prediction or the propensity score estimation is accurate, the overall estimation remains reliable. This robustness is a key strength of our method.
> * **Model architecture**: The shared representation space utilized in our model is a widely adopted approach in treatment effect estimation and multi-task learning [1,2,3]. By employing multi-head structures and multi-task loss functions, the model is able to capture all necessary information within the representation space. One significant advantage of this design is improved sample efficiency: while we may lack sufficient samples from the target distribution, we have ample samples from the training distribution. This allows the model to learn better representations from the available data.
> * **Experiments**: Furthermore, as illustrated in Appendix D.4 (Figure 7), our model demonstrates superior prediction accuracy for both outcome prediction and propensity score estimation compared to single-network architectures.
>
> [1] Nonparametric Estimation of Heterogeneous Treatment Effects: From Theory to Learning Algorithms. Alicia Curth, Mihaela van der Schaar. AISTATS 2021
>
> [2] Adapting Neural Networks for the Estimation of Treatment Effects. Claudia Shi, David M. Blei, and Victor Veitch. NeurIPS 2019.
>
> [3] Learning Shared Representations for Value Functions in Multi-task Reinforcement Learning. Diana Borsa, Thore Graepel, John Shawe-Taylor.
>
>
> ### **Q3. How much risk assessment would be to clinicians**
>  Thank you for raising this thoughtful question. We deeply appreciate your consideration of the practical implications of our work.
> - Our attribution results are designed to provide a comprehensive framework for both data scientists and clinicians to understand the distribution shifts responsible for performance drops. Specifically, the interpretability of both static features and extracted non-stationary properties allows clinicians to identify the underlying reasons behind model failures. This can facilitate actionable insights in medical practice.
> - Moreover, our safe and risk region analysis further enhances practical utility by enabling informed decision-making in deployment scenarios. Even in cases where the model cannot be modified, this analysis supports smarter and safer deployment strategies in real-world settings.
>
> ### **Q4. Illustration of Equation (39)**
> Equation (39) relies on the assumption that the samples are independent, which is a standard assumption in similar settings. In our work, each sample represents the non-stationary properties extracted from the time-series data of a single patient, and we assume that these samples are independent.
>
> To ensure clarity, we will explicitly state this assumption in the main body of the paper. We appreciate your suggestion and will make the necessary adjustments to improve transparency.

---

> > ### Author Response · Authors · 2024-11-22
> >
> > Dear Reviewer 36tU
> >
> > We are sincerely grateful for your time and energy in the review process. Thank you so much for your support!
> >
> > We hope that our responses have been helpful. Given the limited time left in the discussion period, please let us know if there are any leftover concerns and if there is anything else we could do to address any further questions or comments.
> >
> > Thank you!
> >
> > Paper Authors

---

> > > ### Comment · Reviewer_36tU · 2024-11-22
> > >
> > > Thank you for the clarifications. Experimental evidence does show that my concerns were unfounded.

---

> > > > ### Author Response · Authors · 2024-11-22
> > > >
> > > > Thank you so much for your support!

---

### Official Review · Reviewer_WGSf · 2024-11-04

**Soundness:** 2
**Presentation:** 2
**Contribution:** 2
**Rating:** 5
**Confidence:** 5

**Summary:**

The paper proposes a data-driven framework to mitigate the distribution shift in time series data. The framework is designed based on characterizing metrics for behavioral properties of time series and attributing the performance degradation to those metrics. Paper provides empirical and theoretical analysis of the proposed method.

**Strengths:**

1) The paper addresses a very important yet underexplored problem. Not many existing works are designed to study and mitigate the distribution shift in time series data, despite the importance of time series models in real-world applications.

2) The paper provides both theoretical and empirical analysis of the proposed method, which makes the model trustworthy and desirable for deployment in real-world applications.

3) The proposed method demonstrates good performance.

**Weaknesses:**

1) The paper is not well-structured. It introduces many ill-defined concepts, without introducing them or justifying what they mean. For instance, what does the paper mean by "shift within shift"?

2) As a follow up to the previous point, the paper also makes several claims which is not justified throughout the paper. For example, the paper claims the the proposed approach analyzes "application-specific" patterns. However, it is unclear why the proposed metrics are application-specific, as they seem to be mostly focused on general properties of time series.

3) In the first part of the framework, the paper introduces four categories of desiderata for time series metrics and proposed different metrics for each category. It is not clear what are the criteria for selecting metrics in each category? Can different metrics be used?

4) The contribution of the paper is incremental to some extent, as it is mainly built upon similar frameworks designed for the static setting[1,2].

[1]Tiffany Tianhui Cai, Hongseok Namkoong, and Steve Yadlowsky. Diagnosing model performance
under distribution shift. arXiv preprint arXiv:2303.02011, 2023

[2]Jean Feng, Harvineet Singh, Fan Xia, Adarsh Subbaswamy, and Alexej Gossmann. A hierarchical
decomposition for explaining ml performance discrepancies. arXiv preprint arXiv:2402.14254,
2024.

**Questions:**

1)  Why is the proposed method based on analyzing "application-specific" patterns? What does the paper mean by "application-specific"?

2) What is the reason behind selecting metrics in each category of desiderata?  Can different metrics be selected?

---

> ### Author Response · Authors · 2024-11-18
>
> Thank you so much for your suggestions to improve the paper's clarity! We would like to address your concerns as follows:
>
> ### **Q1. Ill-defined/Unclear concepts**:
> Thank you for your thoughtful suggestions to improve the clarity of our paper. Below, we clarify key definitions and claims:
> * Shift within shift: This term distinguishes the type of distribution shifts we address from the natural temporal shifts inherent in time-series data (e.g., changes in features over time). Specifically, the distribution shifts discussed in our work extend beyond static settings to include dynamic changes in the non-stationary properties of time-series data, such as trends, seasonality, and frequency. To improve the clarity, we will change the term to ***shift in non-stationary properties***.
> * Application-specific: The key idea of this work is to understand the specific distribution shift patterns first when facing performance drop in real applications. This concept focuses on the nature of the problem itself, rather than being tied to a specific data modality (e.g., time-series). To enhance clarity and avoid potential misunderstanding, we will revise the term from "application-specific" to "***problem-specific***".
>
> We have incorporated this into the main body, and please feel free to point out other points that you’d specifically like us to fix for better clarity.
>
>
> ### **Q2. Criteria for selecting metrics in each category**
> For each category, we collect metrics from various domains including statistics, analytical chemistry, signal processing, finance, etc. Since this is the first work to deal with this,  we collect as many as possible metrics to improve the attribution power. We acknowledge that these metrics are not exhaustive, and more metrics can be added (as said in line 188).
>
> ### **Q3. Contribution is incremental**
> We would like to clarify that our contributions are significant in problem setting level, methodology level, and theory level:
> * **Setting level**: We focus on time-series settings, where we mainly consider the shift in non-stationary properties. This setting is ***much more complicated***, and we are the first to deal with this. Previous methods cannot deal with this effectively. As shown in Appendix D.3 (Table 2), static region analysis cannot produce reliable safe/risk regions.
> * **Methodology level**: Our attribution method is ***novel and different from previous ones***.
> We formulate the attribution problem as an average treatment effect estimation problem, marking, to the best of our knowledge, the first instance of this approach. And we derive the doubly robust estimator to estimate the attribution score.
> Tiffany’s work [1] uses inverse propensity score reweighting to estimate the score, while our estimator is doubly robust (contain hers). That is, when their method works, our estimator works too.
> Jean’s work [2] uses Shapley value for the attribution, which estimates the average effect of one feature on all other possible feature subsets. However, as discussed in Appendix C, we focus on the impact of one single data property on the model’s performance degradation, while keeping all other properties (or features) constant. As a result, a full calculation of Shapley values may be unnecessary and less suitable for our objectives.
> * **Theory level**: Our attribution has a solid theoretical foundation. Rather than simply incorporate average treatment effect estimation, we ***rigorously prove the unconfoundedness assumption within our framework—a substantial advancement over traditional causal inference literature***, where this assumption is typically taken for granted. Based on this, we prove the double robustness of our estimator. Our framework thus paves the way for a novel perspective on performance attribution, creating opportunities for further exploration in this domain.
>
> [1]Tiffany Tianhui Cai, Hongseok Namkoong, and Steve Yadlowsky. Diagnosing model performance under distribution shift. arXiv preprint arXiv:2303.02011, 2023
>
> [2]Jean Feng, Harvineet Singh, Fan Xia, Adarsh Subbaswamy, and Alexej Gossmann. A hierarchical decomposition for explaining ml performance discrepancies. arXiv preprint arXiv:2402.14254, 2024.

---

> > ### Author Response · Authors · 2024-11-22
> >
> > Dear Reviewer WGSf
> >
> > We are sincerely grateful for your time and efforts in the review process.
> >
> > We hope that our responses have been helpful. Given the limited time left in the discussion period, please let us know if there are any leftover concerns and if there is anything else we could do to address any further questions or comments. We really look forward to your reply.
> >
> > Thank you!
> >
> > Paper Authors

---

> > > ### Comment · Reviewer_WGSf · 2024-11-23
> > > **Response to the Authors**
> > >
> > > I'd like to thank the authors for their response and clarifications. I appreciate their effort in improving the clarity of the paper.
> > >
> > > However, my major concern about this paper regarding *the lack of novelty* (as also noted by reviewer WKbg) and the *limited applicability of the method*, primarily due to the lack of guidance and transparency in the selection criteria, remains. I highly suggest that the paper provides more information regarding how the metrics have been selected, making it feasible for others to extend this method with the criteria for their applications. Due to aforementioned reasons, I would like to keep my score unchanged.

---

> > > > ### Author Response · Authors · 2024-11-24
> > > >
> > > > Thank you very much for your reply!
> > > >
> > > > We wish to clarify the steps we have taken to provide clear guidance and transparency in metric selection, along 3 dimensions as we believe there might be a misunderstanding.
> > > >
> > > > * **1. Desiderata for metrics**:
> > > > To provide a principled and transparent basis for metric selection, we introduce desiderata for the types of metrics (Section 3.1).
> > > > Specifically, our desiderata define the properties that metrics should capture in time-series data, grouped into four categories:
> > > >      * **Global Characteristics**: Capturing long-term trends, averages, and periodicities.
> > > >      * **Local Dynamics**: Measuring short-term variability and anomalies.
> > > >      * **Structural Changes**: Identifying abrupt shifts in data generation processes.
> > > >      * **Multivariate (Inter-Series) Relationships:** Quantifying dependencies between multiple time series.
> > > >
> > > > By organizing metrics into these categories, we offer a structured framework to guide practitioners in selecting or tailoring metrics for their specific datasets.
> > > >
> > > > *  **2. Sufficiency Measure**:
> > > > To systematically assess the quality and relevance of metrics, we employ a sufficiency measure, which quantifies the predictive power of selected metrics in explaining performance degradation. As theoretically demonstrated by Proposition 2, metrics with better sufficiency contribute enable more reliable attribution. We believe such a measure provides a data-driven measure to evaluate & refine metric selection in addition.
> > > >
> > > > * **3. Additional Metrics or extensions**
> > > > While we acknowledge that our current set of metrics is not exhaustive (L188), TSSA as a framework permits users to extend it with additional metrics that align with the defined desiderata. By linking sufficiency to predictive performance, this then ensures that newly added metrics have utility for attribution.
> > > >
> > > > Thank you for helping us ensure the above 3 dimensions are highlighted as clearly as possible to improve the paper!
> > > >
> > > > We really hope this clarification addresses the your concerns around metrics, for both metric selection and extension.

---

> > > > > ### Author Response · Authors · 2024-11-25
> > > > >
> > > > > Dear Reviewer WGSf,
> > > > >
> > > > > We sincerely appreciate your thoughtful feedback, which has provided valuable guidance for enhancing the clarity and transparency of our metric selection.
> > > > >
> > > > > We are pleased to have had the opportunity to clarify the three dimensions in our previous response, which we have now included in Appendix F.1. We believe these updates will be helpful for others using the method. We understand that the current score of 5 was based on an earlier version of the manuscript. In light of the substantial updates and clarifications—driven by your insightful feedback—we respectfully request that you reconsider your rating. We hope you will agree that these revisions have significantly improved the quality of the paper.
> > > > >
> > > > > Thank you once again for your invaluable guidance, which has greatly contributed to the improvement of our work.
> > > > >
> > > > > Best regards,
> > > > >
> > > > > Paper Authors

---

### Meta-Review · Area_Chair_NBFS · 2024-12-20

**Metareview:**

**Summary**: The paper proposes a new method for attributing model performance differences to distribution shifts relevant to a time-series setting. The method primarily builds on attribution methods proposed in static setting by extending to non-stationary settings and improving statistical estimation properties of the estimators (previous work did not focus on overcoming overlap issues using doubly robust estimators). Building on standard theory on doubly robust estimation, the authors provide consistency results for their estimators. Empirical evaluation consists of real world time-series data including healthcare data (MIMIC-III, though unclear why MIMIC-IV isn't used). The interpretation of the shifts also builds a bit on prior work (Figure 3 was proposed in Liu et al 2023) to understand "safety" and "risky" regions of the feature space where performance drop may be higher.

**Strengths**:
1. Reviewers have found the contributions interesting and worthwhile, especially for time-series settings
2. Reasonable empirical performance is demonstrated in real-world data
3. Reviewers find the doubly robust estimator of attributions and the corresponding theory to be a strong contribution

**Weaknesses**:
1. Paper does not seem comprehensive with respect to prior work and has missed many important citations and comparisons. For example some where not highlighted by the reviewers [1, 2, 3], though the focus is on attributions of performance changes as opposed to sensitivity analysis directly, nonetheless, the comparison in empirical evaluation is therefore weak.
2. Overall the contribution is still incremental considering it heavily builds on Liu et al and Cui et al.'s work on attributions to distribution shifts, including the analysis of predicting regions where shifts are likely to result in higher drops.
3. The terminology and writing is a bit ad-hoc, but some were addressed, for example changing "shifts in shifts" (which is definitively not precise) to "shifts in non-stationary properties".

**Justification**:
In my opinion the contribution is fairly incremental and the most informed reviewer in the area roughly agrees with the assessment. Considering that all other reviewers have found the contributions important, and the authors have addressed most of the concerns raised by reviewers, I am recommending an accept. I urge the authors to be a bit more comprehensive in their literature review.

[1] Harini Suresh, Nathan Hunt, Alistair Johnson, Leo Anthony Celi, Peter Szolovits, Marzyeh Ghassemi Proceedings of the 2nd Machine Learning for Healthcare Conference, PMLR 68:322-337, 2017.
[2] Tonekaboni, Sana, et al. "What went wrong and when? Instance-wise feature importance for time-series black-box models." Advances in Neural Information Processing Systems 33 (2020): 799-809.
[3] Choi, Edward, Mohammad Taha Bahadori, Jimeng Sun, Joshua Kulas, Andy Schuetz, and Walter Stewart. "Retain: An interpretable predictive model for healthcare using reverse time attention mechanism." Advances in neural information processing systems 29 (2016).

**Additional Comments On Reviewer Discussion:**

No additional issues were raised in discussion.

---

### Decision · Program_Chairs · 2025-01-22

Accept (Poster)